

# A General Model of Radial Dispersion with Wellbore Mixing and Skin Effect

Wenguang Shi[1], Quanrong Wang[1,2,3*], Hongbin Zhan[4] and Renjie Zhou[5]

[1]School of Environmental Studies, China University of Geosciences, 388 Lumo Road, Wuhan 430074, China

[2]State Environmental Protection Key Laboratory of Source Apportionment and Control of Aquatic Pollution, Ministry of Ecology and Environment, Wuhan, Hubei 430074, PR China

[3]Hubei Key Laboratory of Yangtze River Basin Environmental Aquatic Science, School of Environmental Studies, China University of Geosciences, Wuhan, Hubei 430074, PR China

[4]Department of Geology and Geophysics, Texas A& M University, College Station, TX 77843-3115, USA

[5]Department of Environmental and Geosciences, Sam Houston State University, Huntsville, TX 77340, USA

*Correspondence to*: Quanrong Wang (wangqr@cug.edu.cn)

## Abstract.

The mechanism of radial dispersion is important for understanding reactive transport in the subsurface and for

estimating aquifer parameters required in the optimization design of remediation strategies. Many previous studies demonstrated that injected solute firstly experienced a mixing process in the injection wellbore, then entered a skin zone after leaving the injection wellbore, and finally moved into the aquifer through advective, diffusive, dispersive, and chemical-biological-radiological processes. In this study, a physically-based new model and associated analytical solutions in Laplace domain are developed by considering the mixing effect, skin effect,

scale effect, aquitard effect and media heterogeneity (in which the solute transport is described in a mobile-immobile framework). This new model is tested against a finite-element numerical model and experimental data. The results demonstrate that the new model performs better than previous models of radial dispersion in interpreting the experimental data. To prioritize the influences of different parameters on the breakthrough curves, a sensitivity analysis is conducted. The results show that the model is sensitive to the mobile porosity and

wellbore volume, and the sensitivity coefficient of wellbore volume increases with the well radius, while it decreases with increasing distance from the wellbore. The new model represents the most recent advancement on radial dispersion study that incorporates a host of important processes that are not taken into consideration in previous investigations.

**Keywords:** Solute transport; Recharge well; Divergent flow; Parameter estimation; Push-and-pull test



# 1 Introduction

Radial dispersion refers to a process of reactive transport under the radial flow condition. One unique feature of radial dispersion (as compared to unilateral dispersion where the flow velocity is unilateral) is that the dispersive transport becomes progressively weaker when the radial distance from the injection/pumping well becomes larger (or the radial flow velocity becomes smaller), thus the relative importance of molecular diffusion (which is assumed to be constant) versus the dispersion becomes progressively stronger with larger radial distance. The radial dispersion problem is both theoretically interesting and practically important in many fields, like chemical engineering (Davis and Davis, 2002), environmental science (Reinhard et al., 1997), and hydrogeology (Webster et al., 1970). Although numerical modelling is probably inevitable and more powerful than the analytical modelling in describing radial dispersion, especially put forward for heterogeneous aquifers with complex initial and boundary conditions, the numerical errors and computational cost are not always trivial issues and have to be considered by the engineers. As an alternative, many analytical models have been developed for radial dispersion around an injection well under rather simplified conditions. Such analytical models can fulfil a host of tasks such as 1) prioritizing the importance of different controlling parameters through a sensitivity analysis; 2) benchmarking the numerical solutions to elucidate the possible numerical errors such as numerical dispersion and artificial oscillation which are notorious for advection-dominated transport problems; 3) providing a quick screening tool before implementing a full-scale comprehensive study.

Because of above-mentioned benefits, significant efforts have been put forward over many decades on developing advanced analytical models of radial dispersion. Some examples include the works of

Hoopes and Harleman (1967), Gelhar and Collins (1971), Tang and Babu (1979), Moench and Ogata (1981), Chen (1985), Chen (1986), Hsieh (1986), Tang and Peaceman (1987), Yates (1988), Falade and Brigham (1989), Chen (1991), Novakowski (1992), Philip (1994), Veling (2001), Huang and Goltz

(2006), Chen et al. (2007), Gao et al. (2009a), Chen et al. (2011), Cihan and Tyner (2011), Veling (2011), Chen et al. (2012), Wang and Zhan (2013a), Hsieh and Yeh (2014), Zhou et al. (2017), Wang et al. (2018), and so on. A general trend of such developments is to provide models that are more robust and can better represent the physical reality. However, despite the enormous efforts up to date, some significant pitfalls still exist and become roadblocks for quick and accurate interpretation of observed

data in the experiments. A major task of this research is to eliminate such pitfalls which are briefly illustrated in the following.

In a well-aquifer system with radial dispersion, the region could be horizontally divided into three parts: wellbore, skin zone, and aquifer formation zone. The skin zone refers to the disturbed region around the well caused by drilling and construction practices or well completion (Yeh and Chang, 2013;Chen et al.,

2012). Correspondingly, the injected solute may experience three processes from the wellbore to the aquifer formation zone.

Firstly, the injected solute goes through a mixing process with native (or pre-injection) water in the wellbore at the early stage of injection, which is called mixing effect. Probably due to the small radius of the well, the mixing effect has been overlooked by almost all the analytical solutions mentioned

above except Novakowski (1992) and Wang et al. (2018), e.g., either by assuming that the well radius was infinitesimal, or assuming that the solute concentration in the wellbore was the same as the concentration of the injected solution (Hoopes and Harleman, 1967;Veling, 2011;Zhou et al., 2017).

Consequently, the solutions developed without considering the wellbore mixing effect may overestimate concentration values in both the wellbore and the aquifer (Novakowski, 1992;Wang et al., 2018). The reason is that the solute concentration in the wellbore is initially zero (when the aquifer is free of solute before the injection), and then increases steadily until it is up to the maximum, which is equal to the concentration of the injected solution.

Secondly, the solute enters the skin zone after leaving the wellbore. Although the dimension of the skin zone might be very small, e.g., ranging from 0.1 m to several meters, numerous previous studies demonstrated that the existence of a skin zone might significantly alter the mechanism of groundwater flow and solute transport (Chen et al., 2012;Hsieh and Yeh, 2014;Yeh and Chang, 2013). This is because the physical properties (such as permeability, porosity, dispersivity, and so on) of the skin zone are often vastly different from their counterparts of the formation zone. Previously, studies on the skin effect were mainly concentrated on the groundwater flow process around the well, and much less attention was paid to solute transport processes. For instance, Chen et al. (2012) and Hsieh and Yeh (2014) are probably the only two studies considering the skin effect among the above-mentioned analytical models on radial dispersion.

Thirdly, the solute moves into the formation zone from the skin zone by advective, diffusive, and dispersive processes. Such processes have been widely described by the traditional advection-dispersion equation (ADE) which is based on Fick's law; however, many recent studies demonstrated that the ADE model mainly worked well for homogeneous (or nearly homogeneous) porous media. As for reactive transport in heterogeneous media, the observed breakthrough curves (BTCs) may exhibit a host of non-Fickian characteristics such as early arrival and heavy tailing (Di Dato et al., 2017;Molinari et al.,



2015). Alternatively, many non-Fickian transport models have been developed, such as the multi-rate

mass transfer model (MRMT), mobile-immobile model (MIM), continuous-time random-walk models

(CTRW), fractional-derivative ADE models (fADE), and so on (Zheng et al., 2019;Lu et al., 2018).

MIM is an extension of ADE by considering both flowing and stagnant regions in porous media and

mass transfer between them (van Genuchten and Wierenga, 1976), and Zhou et al. (2017) derived the

MIM solutions associated with radial dispersion. As for the MRMT, CTRW, and fADE models, the

analytical solutions associated with radial dispersion are usually unavailable or difficult to develop.

Meanwhile, these theories are usually not easy to apply for solving regional-scale transport problems, as

pointed out in a recent study (Zheng et al., 2019). Besides the MRMT, MIM, CTRW, and fADE models,

another approach to represent the heterogeneity is to use a scale-dependent dispersivity (or dispersion)

in the ADE or MIM models (Haddad et al., 2015;Gelhar et al., 1992). Gao et al. (2009a) and Chen et al.

(2007) discussed radial dispersion and found that the scale-dependent dispersion effect was not

negligible.

The differences among the currently available analytical solutions for radial dispersion have been

reviewed and summarized in Table 1. As one can see from this table, the mixing effect in the wellbore

was ignored in all of the models except for Novakowski (1992) and Wang et al. (2018). Only Chen et al.

(2012) and Hsieh and Yeh (2014) took the skin effect into account. The differences among the solutions

of Tang and Babu (1979), Moench and Ogata (1981), Hsieh (1986), Tang and Peaceman (1987), Yates

(1988), Cihan and Tyner (2011), and Chen et al. (2012a) mainly consist in the boundary conditions,

source-injection types (instantaneous or continuous), and initial conditions.

In summary, no existing analytical model has ever considered the mixing effect, skin effect, and media

heterogeneity effect (which is described using MIM) simultaneously. Although the numerical method is

more powerful than the analytical method for problems with complex initial and boundary conditions

and heterogeneous aquifers of interest, numerical errors could not be avoided easily for the MIM

models of concern here, such as numerical dispersion and numerical oscillation issues (Zheng and

Wang, 1999;Wang and Zhan, 2013b). Meanwhile, the analytical solutions are usually computationally

more efficient than the numerical solutions, and can be easily coupled into optimization algorithms for

problems related to parameter estimation (Neuman and Mishra, 2012). Therefore, a primary purpose of

this study is to develop such an analytical model. Furthermore, the accuracy and robustness of the

developed model will be tested against a finite-element numerical simulation and experimental data.

Moreover, a sensitivity analysis will be conducted to prioritize the influences of various controlling

parameters on the newly developed radial dispersion reactive transport model.

## 2 Methods

### 2.1 Mathematical model of radial dispersion

An aquifer is assumed to be confined, homogeneous, horizontally isotropic, with a constant thickness,

and fully penetrated by a well from which the solute is injected. A cylindrical coordinate system is

established with the $r$-axis horizontal and the $z$-axis vertically upward. The origin of the coordinate

system is located at intersect of the well center and the middle elevation of the aquifer. A schematic

diagram of the problem is available in Figure S1 of *Supplementary Materials*.





In this study, we mainly focus on developing analytical solutions of radial dispersion with a Heaviside step source (or step function for abbreviation hereinafter), as solutions of a variety of injection scenarios

can be easily obtained on the basis of such a step source solution. Assuming that $t_{inj}$ is the duration of the step source, the solute source concentration ($C_0$) is nonzero when time is smaller than $t_{inj}$, while it is zero when time is greater than $t_{inj}$. The flow rate is constant in the entire time smaller than $t_{inj}$. When $t_{inj}$ approaches zero but the total injected mass remains finite, the model of the step source reduces to the model of the instantaneous injection. Similarly, the model of the step source becomes the

model of the continuous injection source when $t_{inj}$ becomes infinity.

Similar to Chen et al. (2012) and Hsieh and Yeh (2014), a two-region (skin and formation) model of radial dispersion is employed to describe the skin effect. In the skin zone, the governing equations of radial dispersion are

$$\theta_{m1}R_{m1}\frac{\partial C_{m1}}{\partial t} = \frac{\theta_{m1}}{r}\frac{\partial}{\partial r}\left(r\alpha_1|v_{a1}|\frac{\partial C_{m1}}{\partial r}\right) - \theta_{m1}v_{a1}\frac{\partial C_{m1}}{\partial r} - \omega_1(C_{m1} - C_{im1}) - \theta_{m1}\mu_{m1}C_{m1}, r_w \leq r \leq r_s, \text{ (1a)}$$

$$\theta_{im1}R_{im1}\frac{\partial C_{im1}}{\partial t} = \omega_1(C_{m1} - C_{im1}) - \theta_{im1}\mu_{im1}C_{im1}, r_w \leq r \leq r_s; \quad \text{(1b)}$$

In the formation zone, one has

$$\theta_{m2}R_{m2}\frac{\partial C_{m2}}{\partial t} = \frac{\theta_{m2}}{r}\frac{\partial}{\partial r}\left(r\alpha_2|v_{a2}|\frac{\partial C_{m2}}{\partial r}\right) - \theta_{m2}v_{a2}\frac{\partial C_{m2}}{\partial r} - \omega_2(C_{m2} - C_{im2}) - \theta_{m2}\mu_{m2}C_{m2}, r > r_s, \text{(1c)}$$

$$\theta_{im2}R_{im2}\frac{\partial C_{im2}}{\partial t} = \omega_2(C_{m2} - C_{im2}) - \theta_{im2}\mu_{im2}C_{im2}, r > r_s, \quad \text{(1d)}$$

where the subscripts ''$m$'' and "$im$" refer to parameters in the mobile and immobile domains,

respectively; the subscripts ''1'' and "2" refer to parameters in the skin and formation regions,





respectively; $C_{m1}$ and $C_{im1}$ are the mobile and immobile concentrations [ML$^{-3}$] of the skin zone, respectively; $C_{m2}$ and $C_{im2}$ are the mobile and immobile concentrations [ML$^{-3}$] of the formation zone, respectively; $r$ is the radial distance [L] from the center of the well; $r_w$ is the well radius; $r_s$ is the radial distance [L] from the center of the well to the outer boundary of the skin zone; $v_a$ is the average radial pore velocity [LT$^{-1}$] in the aquifer; $v_{a1} = \frac{u_{a1}}{\theta_{m1}}$ ; $v_{a2} = \frac{u_{a2}}{\theta_{m2}}$; $u_{a1}$ and $u_{a2}$ represent Darcian velocities [LT$^{-1}$] in the skin and formation zones, respectively; $\alpha_1$ and $\alpha_2$ represent the longitudinal dispersivities [L] in the skin and formation zones, respectively; $\mu_{m1}, \mu_{im1}, \mu_{m2}$ and $\mu_{im2}$ are reaction rates for the first-order reaction rate, or the first-order biodegradation, or the radioactive decay [T$^{-1}$]; $\theta_{m1}, \theta_{im1}, \theta_{m2}$ and $\theta_{im2}$ are porosities; $R_{m1}, R_{im1}, R_{m2}$ and $R_{im2}$ are retardation factors [dimensionless]; $\omega_1$ and $\omega_2$ represent the first-order mass transfer coefficients [T$^{-1}$] between the mobile and immobile dissolved phases in the skin and formation zones, respectively. One point to note is that the molecular diffusive effect is assumed to be negligible in above governing equations.

Assuming that the skin and formation zone are initially free of solute, the initial conditions are

$$C_{m1}(r,t)|_{t=0} = C_{im1}(r,t)|_{t=0} = C_{m2}(r,t)|_{t=0} = C_{im2}(r,t)|_{t=0} = 0, r \geq r_w. \tag{2}$$

The outer boundary condition at an infinite distance is

$$C_{m2}(r,t)|_{r\to\infty} = C_{im2}(r,t)|_{r\to\infty} = 0. \tag{3}$$

Two types of models have been widely applied to the boundary condition at the well screen: the mass flux continuity (MFC)model and the resident concentration continuity (RCC) model. The RCC model is

$$[C_{m1}(r,t)]|_{r=r_w} = \left[C_{inj}(t)\right]\Big|_{r=r_w}, 0 < t \leq t_{inj}, \tag{4a}$$





$[C_{m1}(r,t)]|_{r=r_w} = [C_{cha}(t)]|_{r=r_w}, t > t_{inj},$ (4b)

and the MFC model is

$$\left[ C_{m1}(r,t) - \alpha_1 \frac{|v_{a1,inj}|}{v_{a1,inj}} \frac{\partial C_{m1}(r,t)}{\partial r} \right]\Bigg|_{r=r_w} = [C_{inj}(t)]|_{r=r_w}, 0 < t \le t_{inj},$$ (5a)

$$\left[ C_{m1}(r,t) - \alpha_1 \frac{|v_{a1,cha}|}{v_{a1,cha}} \frac{\partial C_{m1}(r,t)}{\partial r} \right]\Bigg|_{r=r_w} = [C_{cha}(t)]|_{r=r_w}, t > t_{inj},$$ (5b)

where $C_{inj}(t)$ and $C_{cha}(t)$ represent the solute concentrations [ML$^{-3}$] in the wellbore before time $t_{inj}$

and after time $t_{inj}$, respectively; $v_{a1,inj}$ and $v_{a1,cha}$ refer to velocities in the injection and chasing

phases, respectively. It was demonstrated that the mass balance requirement could not be satisfied in the

RCC model, while the resident concentration was not continuous in the MFC model (Wang et al. 2018).

Many experimental studies demonstrated that the MFC model performed better than the RCC model

(Novakowski, 1992a). Therefore, the MFC model will be used to describe the boundary condition in the

wellbore in this study. Comparing Eqs. (4) and (5), one may find that the main difference between these

two models is whether the dispersivity is involved or not. Recently, Wang et al. (2019) pointed out that

the conflicts between these two models could be resolved by a scale-dependent dispersivity, which was

zero at the well screen, and increased with the travel distance of solute. This is because when the

dispersivity is zero in Eqs. (5a) - (5b), the MFC model reduces to the RCC model. The model of the

scale-dependent dispersivity will be discussed in Section 2.4.

When taking into account the mixing effect in the wellbore, one has

$$V_{w,inj} \frac{dC_{inj}}{dt} = -\xi v_{a1,inj}(r_w)[C_{inj}(t) - C_0], 0 < t \le t_{inj},$$ (6)





$$V_{w,cha} \frac{dC_{cha}}{dt} = -\xi v_{a1,cha}(r_w)[C_{cha}(t)], t > t_{inj}, \tag{7}$$

where $V_{w,inj}$ is the volume [L$^3$] of water in the wellbore when $t \le t_{inj}$, and $V_{w,inj} = \pi r_w^2 h_{w,inj}$; $h_{w,inj}$

is the water level [L] in the wellbore when $t \le t_{inj}$; $\xi = 2\pi r_w \theta_{m1} B$; $B$ is thickness [L] of

aquifer; $V_{w,cha}$ is the volume [L$^3$] of water in the wellbore when $t > t_{inj}$, and $V_{w,cha} = \pi r_w^2 h_{w,cha}$;

$h_{w,cha}$ is the water level [L] in the wellbore when $t > t_{inj}$; $v_{a1,inj}(r_w)$ is velocity at the well screen in

the injection phase, and $v_{a1,inj}(r_w) = \frac{Q_{inj}}{2\pi B r_w \theta_{m1}}$; $v_{a1,cha}(r_w)$ is velocity at the well screen in the chasing

phase, and it equals to $\frac{Q_{cha}}{2\pi B r_w \theta_{m1}}$; $Q_{inj}$ and $Q_{cha}$ are the well flow rates [L$^3$T$^{-1}$] in the injection and

chasing phases, respectively. The mass balance for the well in Eqs. (6) -(7) is only relevant when

velocity is greater than zero, because it does not contain terms for possible diffusive losses.

The water level in the wellbore (e.g., $h_{w,inj}$ and $h_{w,cha}$ ) could be determined by solving the

groundwater flow model. In the steady state, one has

$$Q_{inj} = 2\pi r B K \frac{dh}{dr}, 0 < t \le t_{inj}, \tag{8}$$

$$Q_{cha} = 2\pi r B K \frac{dh}{dr}, t > t_{inj}, \tag{9}$$

where $K$ is the hydraulic conductivity [LT$^{-1}$], and $K = \begin{cases} K_1 & \text{when } r_w \le r < r_s \\ K_2 & \text{when } r_s \le r \end{cases}$; $K_1$ and $K_2$ are the

hydraulic conductivities [LT$^{-1}$] of skin and formation zones, respectively.

By respectively conducting the integration on Eq. (8) and Eq. (9) from $r_w$ to $r_s$ and from $r_s$ to $r_e$, the

water level in the wellbore could be obtained as follows





$$h_{w,inj} = h_0 + \frac{Q_{inj}}{2\pi BK_1} ln \frac{r_s}{r_w} + \frac{Q_{inj}}{2\pi BK_2} ln \frac{r_e}{r_s}, 0 < t \le t_{inj}, \tag{10}$$

$$h_{w,cha} = h_0 + \frac{Q_{cha}}{2\pi BK_1} ln \frac{r_s}{r_w} + \frac{Q_{cha}}{2\pi BK_2} ln \frac{r_e}{r_s}, t > t_{inj}, \tag{11}$$

where $r_e$ is the radial distance [L] from the center of the well to the outer boundary of the formation

zone; $h_0$ is hydraulic head [L] at $r_e$. One could find that a finite radius $r_e$ is needed to keep $h_w$ finite. It

seems contradicts with the boundary condition of the transport problems, which is at the infinity as

shown in Eq. (3). The reason for such a "contradiction" could be explained as follows. In reality, the

influence area is limited by the finite injection rate and the finite injection time of the well (from a plane

view perspective), bounded by a circle with a radius of $r_e$ where the hydraulic head is almost constant

and the flow velocity is almost zero.

At the interface between the skin and formation zones, the concentration and dispersive flux have to be

continuous, and one has

$$C_{m1}(r_s, t) = C_{m2}(r_s, t), t > 0, \tag{12}$$

$$\left[ \alpha_1 |v_{a1}| \frac{\partial C_{m1}(r,t)}{\partial r} \right]\Big|_{r=r_s} = \left[ \alpha_2 |v_{a2}| \frac{\partial C_{m2}(r,t)}{\partial r} \right]\Big|_{r=r_s}, t > 0. \tag{13}$$

Since Darcy fluxes (advective solute fluxes) are continuous, it follows that dispersive fluxes have to be

continuous, which is Eq. (13).

Here, it is worthwhile to comment on the nature of using the MIM approach to describe transport in

heterogeneous aquifers. First, it has been commonly observed that the aquifer heterogeneity renders the

use of ADE invalid in many cases as ADE is developed and used primarily for homogeneous aquifers.

In particular, ADE fails to explain the early breakthrough and long tailing phenomena that are

frequently observed in transport in heterogeneous aquifers, as illustrated in the introduction. Second, a

striking feature of a heterogeneous aquifer is that a sequence of mobile and less mobile regions co-exist while a homogeneous aquifer may be simplified as a single (mobile) region. Ideally, if one knows exactly the spatial distribution of those mobile and less mobile regions and their associated flow and transport parameters, one will be able to use a high-resolution numerical simulator to predict the flow and transport process precisely. Unfortunately, this is not feasible for most practical cases. Therefore, as

an alternative, we have adopted the concept of MIM approach in which two continuums consisting of a mobile domain and an immobile domain co-exist over the entire heterogeneous aquifer. Each of these two continuums itself has uniform flow and transport parameters (such as porosity, retardation factor, etc.) for the sake of simplification. Furthermore, mass can transfer between these two continuums in a certain fashion, usually using the first-order rate-limited equation. Third, this alternative approach has

successfully explained a number of phenomena that cannot be explained using ADE, for instance, the early breakthrough and long tailing issues. Later on, the two-continuum MIM approach has been expanded to multiple-continuum MIM approach, or namely the multi-rate MIM approach to even better capture the transport features in a heterogeneous aquifer (Vangenuchten and Wierenga, 1976;Elenius and Abriola, 2019). In summary, the use of MIM is not to incorporate the spatial variation of flow and

transport parameters that are mostly unknown. Instead, it is based on an alternative approach, using two or more interrelated continuums, and in each continuum the flow and transport parameters remain uniform over space. To date, the validation of the MIM model has been tested by numerous experimental studies (Griffioen et al., 1998;Gao et al., 2009b;Elenius and Abriola, 2019).





## 2.2 Solution of radial dispersion

In this study, dimensionless forms of parameters used in the derivation of analytical solution are defined

as: $C_{m1D} = \frac{C_{m1}}{C_0}$, $C_{im1D} = \frac{C_{im1}}{C_0}$, $C_{m2D} = \frac{C_{m2}}{C_0}$, $C_{im2D} = \frac{C_{im2}}{C_0}$, $C_{inj,D} = \frac{C_{inj}}{C_0}$, $C_{cha,D} = \frac{C_{cha}}{C_0}$, $t_D = \frac{|A|t}{\alpha_2^2 R_{m1}}$,

$t_{inj,D} = \frac{|A|t_{inj}}{\alpha_2^2 R_{m1}}$, $r_D = \frac{r}{\alpha_2}$, $r_{wD} = \frac{r_w}{\alpha_2}$, $r_{sD} = \frac{r_s}{\alpha_2}$, $r_{0D} = \frac{r_0}{\alpha_2}$, $\mu_{m1D} = \frac{\alpha_2^2 \mu_{m1}}{A}$, $\mu_{im1D} = \frac{\alpha_2^2 R_{m1} \mu_{im1}}{R_{im1} A}$, $\mu_{m2D} =$

$\frac{\alpha_2^2 \mu_{m2} R_{m1}}{A R_{m2}}$, $\mu_{im2D} = \frac{\alpha_2^2 R_{m1} \mu_{im2}}{R_{im2} A}$ and $A = \frac{Q}{2\pi B \theta_{m1}}$.

The detailed derivation of the analytical solution in the Laplace domain could be seen in Section S1 of

*Supplementary Materials*. The analytical solution is

$$\bar{C}_{m1D} = N_1 \exp\left(\frac{r_D}{2\lambda}\right) A_i(y_1) + N_2 \exp\left(\frac{r_D}{2\lambda}\right) B_i(y_1), r_{wD} \leq r_D \leq r_{sD}, \tag{14a}$$

$$\bar{C}_{im1D} = \frac{\varepsilon_{im1}}{s + \varepsilon_{im1} + \mu_{im1D}} \bar{C}_{m1D}, r_{wD} \leq r_D \leq r_{sD}, \tag{14b}$$

$$\bar{C}_{m2D} = N_3 \exp\left(\frac{r_D}{2}\right) A_i(y_2), r_D > r_{sD}, \tag{15a}$$

$$\bar{C}_{im2D} = \frac{\varepsilon_{im2}}{s + \varepsilon_{im2} + \mu_{im2D}} \bar{C}_{m2D}, r_D > r_{sD}, \tag{15b}$$

where $A_i(\cdot)$ and $B_i(\cdot)$ are the Airy functions of the first kind and second kind, respectively; $C_{inj,D}$ and

$C_{cha,D}$ could be determined by Eqs. (A18) - (A19), which can be seen in Section S1 of *Supplementary*

*Materials*; $A_i'(\cdot)$ and $B_i'(\cdot)$ are the derivatives of the Airy function of the first kind and second kind,

respectively; $\lambda = \frac{\alpha_1}{\alpha_2}$, $\eta = \frac{\theta_{m1} R_{m1}}{\theta_{m2} R_{m2}}$, $y_1 = \left(\frac{E_1}{\lambda}\right)^{1/3} \left(r_D + \frac{1}{4\lambda E_1}\right)$, $y_{1s} = \left(\frac{E_1}{\lambda}\right)^{1/3} \left(r_{sD} + \frac{1}{4\lambda E_1}\right)$, $y_2 =$

$(E_2)^{1/3} \left(r_D + \frac{1}{4E_2}\right)$, $y_{2s} = (E_2)^{1/3} \left(r_{sD} + \frac{1}{4E_2}\right)$, $E_1 = s + \varepsilon_{m1} + \mu_{m1D} - \frac{\varepsilon_{m1} \varepsilon_{im1}}{s + \varepsilon_{im1} + \mu_{im1D}}$, $E_2 =$





$\quad \frac{1}{\eta}\left(s + \varepsilon_{m2} + \mu_{m2D} - \frac{\varepsilon_{m2}\varepsilon_{im2}}{s+\varepsilon_{im2}+\mu_{im2D}}\right)$, $\beta_{inj} = \frac{V_{w,inj}r_{wD}}{\xi R_{m1}\alpha_2}$, $\beta_{cha} = \frac{V_{w,cha}r_{wD}}{\xi R_{m1}\alpha_2}$, $\varepsilon_{m1} = \frac{\omega_1\alpha_2^2}{A\theta_{m1}}$, $\varepsilon_{im1} = \frac{\omega_1\alpha_2^2 R_{m1}}{A\theta_{im1}R_{im1}}$,

$\varepsilon_{m2} = \frac{\omega_2\alpha_2^2 R_{m1}}{A\theta_{m2}R_{m2}}$ and $\varepsilon_{im2} = \frac{\omega_2\alpha_2^2 R_{m1}}{A\theta_{im2}R_{im2}}$; $s$ is the dimensionless Laplace transform parameter in respect to

dimensionless time $t_D$. the expressions for $N_1$, $N_2$ and $N_3$ are listed in Table 2.

From Eqs. (14) - (15), one may find that it is not easy to analytically invert the Laplace-domain solution

to obtain the real-time solution. Alternatively, numerical Laplace transform techniques such as the

Fourier series method, Zakian method, Schapery method, de Hoog method, Stehfest method are called

in, where the de Hoog and Stehfest methods perform better for problems related to radial dispersion

(Wang and Zhan, 2015). In this study, the de Hoog method will be employed to conduct the inverse

Laplace transform.

### 2.3. Special cases of the new solution

The new solution of this study considers the mixing effect, skin effect, and media heterogeneity (which

is described using MIM) simultaneously, and the solute is injected into the well as a step source. This

general solution encompasses many previous studies as special cases. For instance, when "$r_s \to \infty$", the

skin effect is excluded; "$t_{inj} \to \infty$" implies that the solute is continuously injected into the well; while

"$t_{inj} \to 0$" means that the solute is instantaneously injected into the well; "$\omega = 0$" implies that the

MIM solution reduces to the ADE solution; "$V_{w,inj} = 0$" or "$r_w = 0$" shows that the mixing effect is

excluded.

Therefore, the new solution reduces to the solutions of Hoopes and Harleman (1967), Gelhar and

Collins (1971), Tang and Babu (1979), Moench and Ogata (1981), Hsieh (1986), Tang and Peaceman





(1987), and Philip (1994) when $r_s \rightarrow \infty$, $t_{inj} \rightarrow \infty$, $\omega = 0$, and $V_{w,inj} = 0$. The solution of Wang et al.

(2018) is a special case of this study when $r_s \rightarrow \infty$, $t_{inj} \rightarrow \infty$, and $\omega = 0$.

## 2.4. Extension of the new solution with scale-dependent dispersivity

Due to the heterogeneities of the porous media, the dispersivity was found to be dependent on travel distance of solute from source, and such phenomenon was firstly observed in the field scale experiment (Dagan, 1988;Gelhar et al., 1992;Pickens and Grisak, 1981a). The field scale effect (i.e., dispersivity

growing with distance from well), is usually considered to be a result of spatial heterogeneity at different scales in the aquifer. Subsequently, the scale-dependent dispersivity phenomenon was also found in controlled laboratory tests, due to heterogeneities caused by the bridging effect and microstructures, although the sediments (as the porous media) are well sorted and carefully packed (Silliman and Simpson, 1987;Berkowitz et al., 2000;Wang et al., 2019;Gao et al., 2010). For example,

Silliman and Simpson (1987) found that the dispersivity continuously increased with distance, based on the experiments conducted in a 2.4×1.07×0.10 m sandbox. Berkowitz et al. (2000) obtained similar conclusions to Silliman and Simpson (1987) in the laboratory-controlled experiment. Wang et al. (2019) also concluded that the scale-dependent model performed better than the scale-independent model in interpreting observed BTCs of the laboratory-controlled experiment. To date, four types of functions

have been widely used to describe the scale-dependent dispersivity, including asymptotic, parabolic, exponential and linear functions, as summarized by Pickens and Grisak (1981b). In this section, the model of the scale-independent dispersivity (e.g., Eqs. (14) - (15) in Section 2) will be extended by considering the linear-asymptotic dispersivity model in the formation zone. As for the other types of





scale-dependent functions, the analytical solutions could be derived using a similar approach. The

formula of the linear distance-dependent dispersivity is

$$\alpha_2(r) = \begin{cases} kr, & r_s \leq r \leq r_0 \\ \alpha_0, & r > r_0 \end{cases},$$ (16)

where $r_0$ is the distance [L] where $\alpha_2(r_0) = \alpha_0$, $k$ is a constant [dimensionless], and the modified

solutions are

$$\bar{C}_{m1D} = \mathcal{T}_1 \, exp\left(\frac{r_D}{2\lambda}\right) A_i(y_1) + \mathcal{T}_2 exp\left(\frac{r_D}{2\lambda}\right) B_i(y_1), r_{wD} \leq r_D \leq r_{sD},$$ (17a)

$$\bar{C}_{im1D} = \frac{\varepsilon_{im1}}{s+\varepsilon_{im1}+\mu_{im1D}} \bar{C}_{m1D}, r_{wD} \leq r_D \leq r_{sD},$$ (17b)

$$\bar{C}_{m2D} = \mathcal{T}_3 r_D^m K_m(\varepsilon_1 r_D) + \mathcal{T}_4 r_D^m I_m(\varepsilon_1 r_D), r_{sD} \leq r_D \leq r_{0D},$$ (18a)

$$\bar{C}_{m2D} = \mathcal{T}_5 \, exp\left(\frac{r_D}{2}\right) A_i(y_3) + \mathcal{T}_6 exp\left(\frac{r_D}{2}\right) B_i(y_3), r_D > r_{0D},$$ (18b)

$$\bar{C}_{im2D} = \frac{\varepsilon_{im2}}{s+\varepsilon_{im2}+\mu_{im2D}} \bar{C}_{m2D}, r_D > r_{sD},$$ (18c)

where $m = \frac{1}{2k}$; $K_m(\cdot)$ is $m^{th}$-order modified Bessel function of the second kind, $I_m(\cdot)$ is $m^{th}$-order

modified Bessel function of the first kind; the expressions for $\mathcal{T}_1, \mathcal{T}_2, \mathcal{T}_3, \mathcal{T}_4, \mathcal{T}_5$ and $\mathcal{T}_6$ are listed in Table

3; $y_3 = (\varepsilon_1)^{1/3}\left(r_D + \frac{1}{4\varepsilon_1}\right)$; $y_4 = (\varepsilon_1)^{1/3}\left(r_{0D} + \frac{1}{4\varepsilon_1}\right)$; $C_{inj,D}$ and $C_{cha,D}$ could be determined by Eqs.

(B15) - (B16), and the detailed derivation of Eqs. (17) - (18) could be seen in Section S2 of

*Supplementary Materials*.

Substituting Eq. (16) into the dispersivity coefficient ($D_\alpha$), one has



$$D_\alpha = \alpha_2 |v_{a1}| + D_0 = \begin{cases} \dfrac{krQ}{2\pi rB\theta_{m1}} + D_0 = \dfrac{kQ}{2\pi B\theta_{m1}} + D_0, \ r_s \le r \le r_0 \\ \dfrac{\alpha_0 Q}{2\pi rB\theta_{m2}} + D_0, \qquad\qquad r > r_0 \end{cases}, \qquad (19)$$

where $D_0$ is molecular diffusion coefficient [$L^2T^{-1}$]. A few interesting features are notable here. First,

because of the unique feature of a divergent flow field in which the velocity is inversely proportional to

the radial distance, and the use of a dispersivity function that is proportional to the radial distance when

$r \le r_0$, the dispersion coefficient in Eq. (19) actually becomes constant. However, one must be aware

that if other types of dispersivity equations are used (such as exponential and parabolic functions), the

dispersion coefficient in Eq. (19) will depend on the radial distance from the well. Second, even when

$D_\alpha$ becomes constant for a linear dispersivity function when $r \le r_0$, the mechanical dispersion is still

dominant, since the value of $D_0$ is generally much smaller than the mechanical dispersion term of

$\dfrac{kQ}{2\pi B\theta_{m1}}$. For instance, the diffusion coefficient in water ranges from $1 \times 10^{-9}$ to $2 \times 10^{-9}$ m$^2$/s, and it is

much smaller in the porous media (Freeze and Cherry 1979). When $k = 0.01$, $Q = 0.1$ m$^3$/s, $B$=1m,

$\theta_{m1} = 0.3$, one has $\dfrac{kQ}{2\pi B\theta_{m1}}$ =5.3 $\times$ $10^{-4}$ m$^2$/s. Therefore, it is reasonable to ignore the molecular

diffusion effect when $r \le r_0$. The values of $\dfrac{kQ}{2\pi B\theta_{m1}}$ is dependent on $k$. The chosen value of $k = 0.01$ is

from experimental studies, for instance, $k = 0.018$ in Chen et al. (2007), and $k = 0.024$ and $0.013$ in

this study as shown in Table 5.

**2.5. Extension of the new solution to a leaky-confined aquifer**

Regardless of Eqs. (14) - (15) or Eqs. (17) - (18), the aquifer is assumed to be completely isolated from

the underlying and overlying aquitards (strictly confined), which might not be true in real applications.

As stated before (Zhan et al., 2009b;Zhan et al., 2009a), it is nearly impossible to maintain a strictly confined condition in terms of transport. That is because as long as solute in the aquifer is in contact

with the upper or lower aquitard, molecular diffusion will always drive the solute from high concentration aquifer into the solute-free aquitard, even if the cross-formation flow in the aquitard does not exist. In fact, such diffusion-driven transport of solute into the aquitard and the subsequent back diffusion (from aquitard to aquifer when the aquifer solute concentration drops below the solute concentration in the aquitards) is responsible for many long tails in aquifer BTCs. The importance of

aquitard in regulating solute transport has indeed been recognized by a number of investigators such as Chen (1985), Chen (1986), Yates (1988), Chen (1991), Novakowski (1992), Wang and Zhan (2013a), and Zhou et al. (2017).

In this section, the solutions of Eqs. (14) - (15) will be extended considering both underlying and overlying aquitards. The detailed derivation of the analytical solution in Laplace domain could be seen

in Section S3 of *Supplementary Materials*.

In the aquifer, the solutions are

$$\bar{C}_{m1D} = T_1 \, exp\left(\frac{r_D}{2\lambda}\right) A_i(\varphi_1) + T_2 exp\left(\frac{r_D}{2}\right) B_i(\varphi_1), r_{wD} \leq r_D \leq r_{sD}, \tag{20a}$$

$$\bar{C}_{im1D} = \frac{\varepsilon_{im1}}{s + \varepsilon_{im1} + \mu_{im1D}} \bar{C}_{m1D}, r_{wD} \leq r_D \leq r_{sD}, \tag{20b}$$

$$\bar{C}_{m2D} = T_3 \, exp\left(\frac{r_D}{2}\right) A_i(\varphi_2), r_D > r_{sD}, \tag{21a}$$

$$\bar{C}_{im2D} = \frac{\varepsilon_{im2}}{s + \varepsilon_{im2} + \mu_{im2D}} \bar{C}_{m2D}, r_D > r_{sD}; \tag{21b}$$

In the aquitards, the solutions are





$$\bar{C}_{umD} = \bar{C}_{m1D} exp\,(a_2 z_D - a_2), r_{wD} \leq r_D \leq r_{sD}, \tag{22a}$$

$$\bar{C}_{umD} = \bar{C}_{m2D} exp\,(a_2 z_D - a_2), r_D > r_{sD}, \tag{22b}$$

$$\bar{C}_{uimD} = \frac{\varepsilon_{uim}}{s + \varepsilon_{uim} + \mu_{uimD}} \bar{C}_{umD},\ r_D > r_{wD}, \tag{22c}$$

$$\bar{C}_{lmD} = \bar{C}_{m1D} exp\,(b_1 z_D + b_1), r_{wD} \leq r_D \leq r_{sD}, \tag{23a}$$

$$\bar{C}_{lmD} = \bar{C}_{m2D} exp\,(b_1 z_D + b_1), r_D > r_{sD}, \tag{23b}$$

$$\bar{C}_{limD} = \frac{\varepsilon_{lim}}{s + \varepsilon_{lim} + \mu_{limD}} \bar{C}_{lmD},\ r_D > r_{wD}, \tag{23c}$$

where letters "$u$" and "$l$" in the subscript represent the upper and lower aquitards, respectively; $\varphi_w = \left(\frac{E_3}{\lambda}\right)^{1/3} \left(r_{wD} + \frac{1}{4\lambda E_3}\right)$, $\varphi_1 = \left(\frac{E_3}{\lambda}\right)^{1/3} \left(r_D + \frac{1}{4\lambda E_3}\right)$, $\varphi_2 = E_4^{1/3} \left(r_D + \frac{1}{4E_4}\right)$; $\varphi_{1s} = \left(\frac{E_3}{\lambda}\right)^{1/3} \left(r_{sD} + \frac{1}{4\lambda E_3}\right)$;

$\varphi_{2s} = E_4^{1/3} \left(r_{sD} + \frac{1}{4E_4}\right)$; the expressions for $a_2, b_1, T_1, T_2$ and $T_3$ are listed in Table 4; $C_{inj,D}$ and $C_{cha,D}$ could be determined by Eqs. (C36) - (C37), which can be seen in Section S3 of *Supplementary Materials*.

The solutions of Chen (1985), Chen (1986), Yates (1988), and Chen (1991) are special cases of this study when $r_s \rightarrow \infty$, $t_{inj} \rightarrow \infty$, $\omega = 0$, and $V_{w,inj} = 0$. When $r_s \rightarrow \infty$, $t_{inj} \rightarrow \infty$, and $V_{w,inj} = 0$, the

new solution reduces to the solution of Zhou et al. (2017). Novakowski (1992) considered the wellbore mixing effect in an aquifer-aquitard system, while he ignored other factors such as the skin effect, scale-dependent dispersivity, and mass transfer between the mobile and immobile domains in porous media.

# 3 Results and discussion

## 3.1 Test of new solutions

To test the new solution of this study, a numerical simulation based on the Galerkin finite-element method is conducted in the COMSOL Multiphysics platform. More details about the numerical simulation setup could be seen in Section S4 of *Supplementary Materials*.

As it is difficult to describe the wellbore mixing effect in COMSOL Multiphysics, the wellbore concentration is computed by the analytical solutions of Eqs. (14) - (15). Figures 1a and 1b show the
comparison of concentration between the numerical and analytical solutions of this study, and good agreement between these two kinds of solutions is evident for different times and locations. The comparisons between the numerical solution and analytical solutions of Eqs. (20) - (23) are shown in Section S4.2 of *Supplementary Materials*, and the agreement is also good between them.

## 3.2 Test of model using experimental data

To test the influence of the mixing effect, skin effect, and heterogeneity of the media on radial dispersion, the experimental data of Chao (1999) is employed. Chao (1999) reported a laboratory experiment of radial dispersion in a sand tank with 244 cm in length, 122 cm in width and 6.35 cm in depth. A well with a radius of 1.0 cm fully penetrated a confined aquifer. Two observation wells were respectively located at 22.5 cm and 30.4 cm away from the well center. Potassium Bromide (KBr) is
chosen as a conservative tracer. Before the tracer is introduced into the wellbore, a steady-state flow field is produced by injecting KBr-free water into the aquifer with a constant injection rate of 9.9 mL/min. The injection time is 5 hours for the tracer while maintaining the same injection rate of 9.9





mL/min. The experimental data of Chao (1999) was interpreted by Gao et al. (2009a) using the model

of Chen et al. (2007), as shown in Figure 2. "SDM" and "CDM" in the legend of Figure 2 refer to the

scale-dependent dispersivity model and the constant dispersivity model, respectively. Chen et al. (2007)

approximated the injection as an instantaneous source (the validity of such a treatment will be addressed

later) and the mass $M$ of the instantaneous injection is calculated by

$$M = C_0 Q_{inj} t_{inj}. \tag{24}$$

The other parameters of the analytical solution are listed in Table 5. The parameters estimated by Gao et

al. (2009a) are also included in Table 5 for comparison. One may find that the goodness-of-fit between

the observed data and models of Gao et al. (2009a) and Chen et al. (2007) seems good at the

observation point close to the well, but they could not capture BTCs at $r$=30.4 cm. This is probably due

to the following two reasons. Firstly, the model of Chen et al. (2007) used to best fit the data is an

instantaneous slug test model, which is a rather gross approximation of the injection which lasted about

5 hours. A more proper way is to treat the 5 hours injection as a step source. Secondly, the solution of

Chen et al. (2007) only considered the scale-dependent dispersivity, but ignored the mixing effect and

the mass transfer between the mobile and immobile domains.

To test the new solutions of this study, we try to best fit the observed data again using the newly

developed model considering the scale-dependent dispersivity, mixing effect and heterogeneity of the

media (described using MIM). As there is no aquitard in the controlled laboratory experiment, the

aquitard effect is irrelevant. Meanwhile, as there is no skin, so the skin effect is not included either. Best





fitness between the analytical solution and the experimental data is an optimization process by minimizing the "error" between them,

$$error = \sum_{i=1}^{N}(C_{OBS} - C_{COM})^2, \tag{25}$$

where $C_{OBS}$ and $C_{COM}$ represent the observed and computed concentrations, respectively; $N$ is the number of sampling points. In this study, the genetic algorithm (GA) is employed to search the optimal parameter values, such as $\theta_{m2}$, $\alpha_1$ and $\omega_1$ for CDM of Eqs. (14) - (15), and $\theta_{m2}$, $\alpha_0$, $k$ and $\omega_1$ for CDM of Eqs. (17) - (18). GA is a stochastic search method, based on natural selection, and it is preferred, due to its efficiency, simple programmability, and robustness. The estimated values of some

key parameters are listed in Table 5. The errors between the observed and computed BTCs by different models are listed in Table 6. Figure 3 shows the fitness between the analytical solution and the experimental data, with and without scale-dependent dispersivity, respectively. As GA converges after 500 generations (iterations), the fitness is good as shown in Figure 3, and the estimated parameters are physically sound.

Comparing Figures 2 and 3 shows that the solutions of this study perform better than the model of Chen et al. (2007), since the fitness is good for both observation locations. To better evaluate the overall performance of the models for both locations, we have used Eq. (25) to compute the errors of best fitness with two BTCs simultaneously in Figures 2 and 3, and the results are listed in the last column of Table 6. This table shows that the new model performs better. For example, when using CDM, the

overall errors for best fitting two locations are 0.89 (which is the summation of 0.06 and 0.83 in Table 6) for Chen et al. (2007), and 0.39 (which is the summation of 0.34 and 0.05 in Table 6) for this study.



When using SDM, however, the overall errors for best fitting two locations are 0.78 for Chen et al. (2007), and 0.25 for this study. Evidently, the model with scale effect is the best choice for interpreting the experimental data.

We have to emphasize that better fitting of one model than the other model with the experimental data should not be used as the only evidence of proof for model performance. That is because a model with more fitting parameters usually performs better than the model with less fitting parameters. Besides the best fitting exercises, however, one should pay more attention to see if the model adequately acknowledges the underlying physiochemical principles governing the transport processes. As far as we

can see, the new model proposed in this study has honored the underlying physiochemical principles governing the radial dispersion process properly. In addition, the model performance (as reflected in the best fitting practice with the experimental data) is also considerably better. Therefore, on the basis of these two considerations, the new model of this study can be regarded as a significant advancement of present knowledge on radial dispersion. Furthermore, the new model is quite general and it

encompasses almost all the existing models as subsets.

### 3.3 Sensitivity analysis

As the new model involves a number of controlling parameters, it is necessary to prioritize the importance of these parameters in terms of their control on the model performance. In this study, a sensitivity analysis involving normalized parameters is conducted as follows (Kabala, 2001;Yang and

Yeh, 2009)

$$SC_{i,j} = I_j \frac{\partial c_i}{\partial I_j}, \tag{26}$$



where $SC_{i,j}$ is the sensitivity coefficient of the $j^{th}$ parameter $I_j$ at the $i^{th}$ time; $C_i$ is the concentration at the $i^{th}$ time. $I_j$ represents any one parameter of interest, like volume of water in the wellbore ($V_w$), $k$, $\theta_m = \theta_{m1} = \theta_{m2}$, $\omega = \omega_1 = \omega_2$, and so on. A larger $\left|SC_{i,j}\right|$ value means the higher sensitivity.

As the expression of the new analytical solution is complex, it is not easy to get the values of $SC_{i,j}$ directly from Eq. (26). Therefore, a finite-difference scheme is used alternatively to approximate the right-hand side term (Kabala, 2001;Yang and Yeh, 2009)

$$SC_{i,j} = I_j \frac{C_i(I_j + \Delta I_j) - C_i(I_j)}{\Delta I_j}, \tag{27}$$

where $\Delta I_j$ is a small increment of $I_j$.

The main parameters of the new model include the volume of the water in the wellbore ($V_w$) for the mixing effect, $r_s$ and $\alpha_s$ for the skin zone, $\theta_m$ and $\omega$ for the MIM model, and $k$ for scale dependent dispersivity. Figures 4a and 4b show $SC_{i,j}$ at $r$=22.5 cm and $r$=30.4 cm, respectively. The parameters used in these two figures are the same as those used in Figure 3.

Two observations could be found from Figures 4a and 4b. Firstly, the results are sensitive to the

parameter of $\theta_m$. To test such a finding, we use the model of this study with the mixing effect to best fit the experimental data of Chao (1999) (shown in Figure S4 in Section S5 of *Supplementary Materials*), and the results show that the influence of the mixing effect could be negligible. Secondly, by comparing Figures 4a and 4b, we find that the sensitivity coefficient of $V_w$, $r_s$ and $\alpha_s$ on BTCs increases with the distance from the wellbore.



Figures 4a and 4b show that the sensitivity coefficient of $V_w$ on BTCs is very small, which might

contradict with the finding reported in some previous studies (Wang et al., 2018). A careful inspection

indicates that the well radius and the initial water level in the wellbore are very small in the experiment

of Chao (1999), resulting in a very small value of $V_w$. From Eqs. (6) - (7), one can see that $V_w$ could be

influenced by the pumping rate, well radius, initial water level ($h_0$), and hydraulic parameters of the

aquifer. In actual field practices, the value of $V_w$ can be significantly larger than what is used in Chao

(1999). Therefore, the sensitivity coefficient of $V_w$ on BTCs will be investigated again using the well

radius and the initial water level that are more commonly seen in field applications, e.g., $r_w$=5.0 cm and

$h_0$=31.75 cm, and the other parameters are the same as ones in Figure 3.

The sensitivity analysis after such modification shows that the parameter with the highest sensitivity

coefficient is still $\theta_m$, but the parameter with the second highest sensitivity coefficient becomes the

volume of water in the wellbore (Figure 5). Figures 6 and 7 illustrate $SC_{i,j}$ of $V_w$ for different $r_w$ and

different observation locations, and that the sensitivity coefficient of $V_w$ increases with the well radius,

but decreases with the distance from the wellbore.

## 4 Conclusions

Radial dispersion is an important process in the fields of chemical engineering, environmental science,

and hydrogeology. It has been commonly employed to describe the reactive transport in the subsurface,

or to estimate aquifer transport parameters (dispersivity, porosity, and reactive rate, etc.) required in



optimization of remediation strategies. However, previous studies did not include all of the mixing

effect, skin effect, and mass transfer between the mobile and immobile domains in porous media.

In this study, a new general model is developed considering all above-mentioned factors. The new

general model is against by a finite-element numerical model and existing experimental data.

Meanwhile, the new model is also expanded considering the effect of the overlying and underlying

aquitards and the scale-dependent dispersivity. The sensitivity analysis is conducted to prioritize

influences of various controlling parameters on BTCs. The following conclusions could be summarized:

(1) The new general model honors the most relevant processes involved in radial dispersion (wellbore

mixing effect, well skin effect, aquitard effect and mass transfer between the mobile and immobile

domains), for which a solution has not yet been presented.

(2) The new general model fits the experimental data of Chao (1999) much better than previous models.

(3) The results are sensitive to parameters $\theta_m$ (mobile porosity) and $V_w$ (the volume of water in the

wellbore). When $V_w$ is very small as in the laboratory experiment of Chao (1999), the sensitivity

coefficient approaches 0. However, for typical values of $V_w$ in actual field applications, the sensitivity

coefficient of $V_w$ increases significantly, and the value is often ranked as the second highest, after that of

$\theta_m$.

(4) The sensitivity coefficient of $V_w$ increases with the well radius, while it decreases with increasing

distance from the wellbore.

**Data availability:** The datasets used and/or analysed during the current study are available from the corresponding author on reasonable request.

**Author contributions:** Methodology, derivation, code, and formal analysis, writing original draft: WS. Conceptualization, writing original draft, writing-review and editing, and supervision: QW. Vetting and technical support: HZ and RZ.

**Competing interests:** The contact author has declared that neither they nor their co-authors have any competing interests.

### Acknowledgments

This research was partially supported by two Programs of the National Natural Science Foundation of China (No.42222702, No.41772252 and No. 41972250); National Key Research and Development Program of China (No. 2021YFA0715900); Innovative Research Groups of the National Nature Science Foundation of China (No. 41521001), the Fundamental Research Funds for Central Universities, China University of Geosciences (Wuhan) (No. CUGGC07); the Natural Science Foundation of Hubei Province (2021CFA089); the 111 Program (State Administration of Foreign Experts Affairs & the Ministry of Education of China, No. B18049); the Belt and Road Special Foundation of the State Key Laboratory of Hydrology-Water Resources and Hydraulic Engineering(No. 2020492011), and Natural Science Foundation of Chongqing (cstc2020jcyj-msxmX1072).



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





# Nomenclature

| Symbol | Description |
|--------|-------------|
| $A_i(\cdot), B_i(\cdot)$ | Airy functions of the first kind and the second kind, respectively |
| $A_i'(\cdot), B_i'(\cdot)$ | Derivative of the Airy functions of the first kind and the second kind, respectively |
| $\alpha_0$ | Longitudinal dispersivity [L] in the formation zone at $r > r_0$ |
| $\alpha_1, \alpha_2$ | Longitudinal dispersivities [L] in the skin and formation zones, respectively |
| $B$ | The thickness [L] of aquifer |
| $b$ | The half of aquifer thickness [L] |
| $C_{m1}, C_{im1}$ | Resident mobile and immobile concentrations [ML$^{-3}$] of the skin zone, respectively |
| $C_{m2}, C_{im2}$ | Resident mobile and immobile concentrations [ML$^{-3}$] of the formation zone, respectively |
| $C_{um}, C_{uim}$ | Resident mobile and immobile concentrations [ML$^{-3}$] of the upper aquitard, respectively |
| $C_{lm}, C_{lim}$ | Resident mobile and immobile concentrations [ML$^{-3}$] of the lower aquitard, respectively |
| $C_{inj}(t),$ $C_{cha}(t)$ | Concentrations [ML$^{-3}$] of tracer in the wellbore at the injection and the chasing phases, respectively |
| $C_0$ | Concentration [ML$^{-3}$] of tracer injected into the wellbore |
| $C_w$ | Concentration [ML$^{-3}$] of tracer in the wellbore |
| $D_u, D_l$ | Vertical dispersion coefficients [L$^2$T$^{-1}$] of the upper and lower aquitards, respectively |
| $D_0$ | Molecular diffusion coefficient [L$^2$T$^{-1}$] |
| $h$ | Hydraulic head [L] |
| $h_0$ | Hydraulic head [L] at the $r_e$ |
| $h_{w,inj}, h_{w,cha}$ | Water level in the wellbore in the injection and chasing phases [L] |
| $k$ | A constant [dimensionless] and ranges from 0 to 1 |
| $K_1, K_2$ | Hydraulic conductivities [LT$^{-1}$] of skin and formation zones, respectively |
| $K_d$ | Equilibrium distribution coefficient [M$^{-1}$L$^3$] for the linear sorption process |
| $I_m(\cdot), K_m(\cdot),$ | The $m^{th}$-order modified Bessel function of the first and second kinds, respectively |
| $Q$ | Pumping rate [L$^3$T$^{-1}$] (negative for injection and positive for pumping) |
| $Q_{inj}, Q_{cha}$ | Well flow rates [L$^3$T$^{-1}$] in the injection and chasing phases, respectively. |
| $r$ | Radial distance [L] from the center of the well |
| $r_s$ | Distance [L] from the center of the well to the outer boundary of the skin zone |
| $r_w$ | Radius [L] of the well |
| $r_e$ | Radial distance [L] from the center of the well to the outer boundary |



| | |
|---|---|
| $r_0$ | Radial distance [L] for the linear distance-dependent dispersivity |
| $R_{m1}, R_{im1}$ | Retardation factors [dimensionless] for the mobile and immobile regions of the skin zone |
| $R_{m2}, R_{im2}$ | Retardation factors [dimensionless] for the mobile and immobile regions of the formation zone |
| $R_{um}, R_{uim}$ | Retardation factors [dimensionless] for the mobile and immobile regions of the upper aquitard |
| $R_{lm}, R_{lim}$ | Retardation factors [dimensionless] for the mobile and immobile regions of the lower aquitard |
| $t$ | Time [T] |
| $t_{inj}, t_{cha}$ | Ending times [T] of the injection and the chasing phases, respectively |
| $v_{a1}, v_{a2}$ | Average radial pore velocities [LT$^{-1}$] of the skin zone, the formation zone, respectively |
| $v_{a1,inj}, v_{a1,cha}$ | Average radial pore velocities [LT$^{-1}$] at the well screen in the injection and chasing phases, respectively. |
| $v_{um}, v_{lm}$ | Vertical velocities [LT$^{-1}$] of the upper and lower aquitards, respectively |
| $\alpha_u, \alpha_l$ | Dispersivities [L] of the upper aquitard and the lower aquitard, respectively |
| $\mu_{m1}, \mu_{im1}$ | Decay constant for radioactive decay or reaction rate coefficient [T$^{-1}$] in the mobile and immobile regions of the skin zone |
| $\mu_{m2}, \mu_{im2}$ | Decay constant [T$^{-1}$] for radioactive decay or reaction rate coefficient in the mobile and immobile regions of the formation zone |
| $\mu_{um}, \mu_{uim}$ | Decay constant [T$^{-1}$] for radioactive decay or reaction rate coefficient in the mobile and immobile regions of the upper aquitard |
| $\mu_{lm}, \mu_{lim}$ | Decay constant [T$^{-1}$] for radioactive decay or reaction rate coefficient in the mobile and immobile regions of the lower aquitard |
| $\theta_{m1}, \theta_{im1}$ | Mobile and immobile porosities [dimensionless] in the skin zone |
| $\theta_{m2}, \theta_{im2}$ | Mobile and immobile porosities [dimensionless] in the formation zone |
| $\theta_{um}, \theta_{uim}$ | Mobile and immobile porosities [dimensionless] in the upper aquitard |
| $\theta_{lm}, \theta_{lim}$ | Mobile and immobile porosities [dimensionless] in the lower aquitard |
| $\rho_b$ | Bulk density [ML$^{-3}$] of the aquifer material |
| $\omega_1, \omega_2$ | First-order mass transfer coefficients [T$^{-1}$] in the skin and formation zones, respectively |
| $s$ | Laplace transform variable with respect to the time $t_D$ |
| **Subscript** | **Description** |
| $D$ | Dimensionless form |
| $m, im$ | Mobile and immobile regions, respectively |
| $inj, cha$ | Injection and chasing phases, respectively |
| $u, l$ | Upper and lower aquitard, respectively |





| 1, 2 | Parameters in the skin and formation regions, respectively |
| --- | --- |
| **Acronyms** | **Description** |
| ADE | Advection-dispersion equation |
| BTCs | The observed breakthrough curves |
| CDM | The constant dispersivity model |
| CTRW | Continuous-time random-walk models |
| fADE | Fractional-derivative ADE models |
| GA | The genetic algorithm |
| MFC | The mass flux continuity |
| MIM | Mobile-immobile model |
| MRMT | The multi-rate mass transfer model |
| RCC | The resident concentration continuity |
| SDM | The scale-dependent dispersivity model |






**Figure Captions**

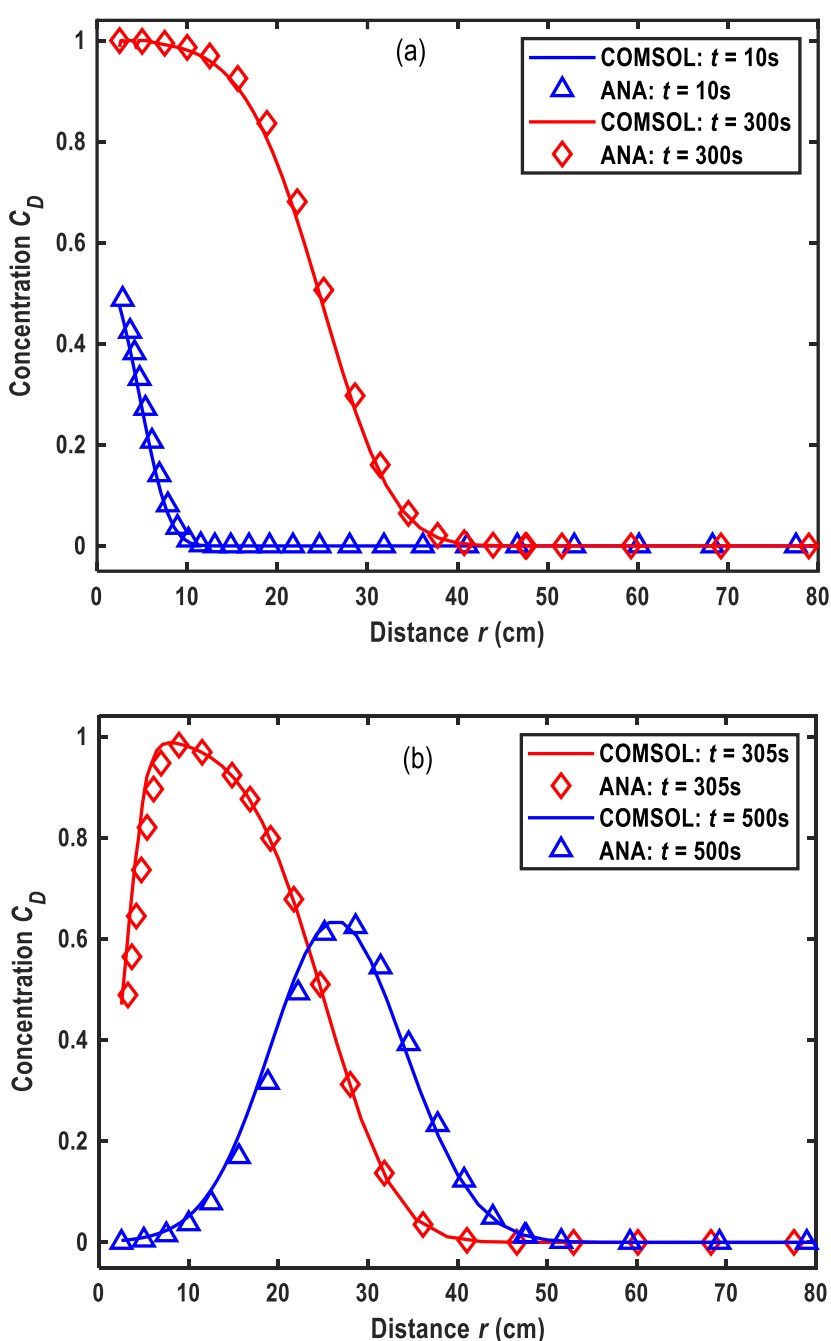

**Figure 1.** Comparison of the numerical solution by COMSOL Multiphysics and the analytical solution
of this study for different times. (a). In the injection phase, (b). In the chasing phase.





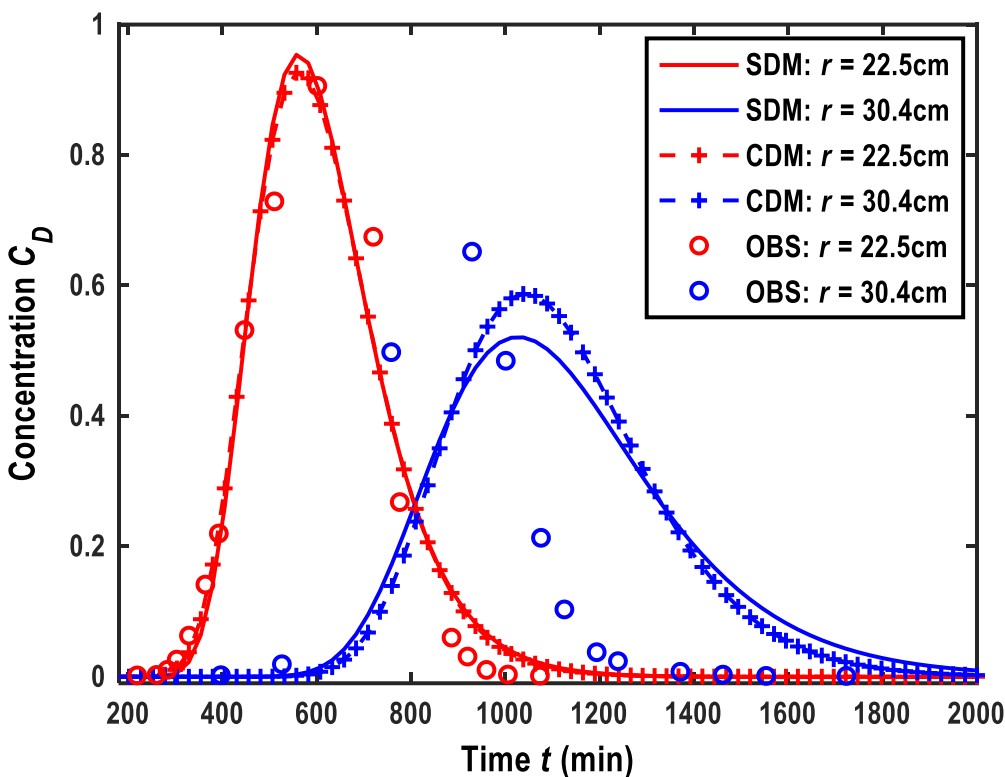

**Figure 2.** Fitness of observed BTC by the solution of Chen et al. (2007) which considers the scale effect but ignores the mixing and skin effects.





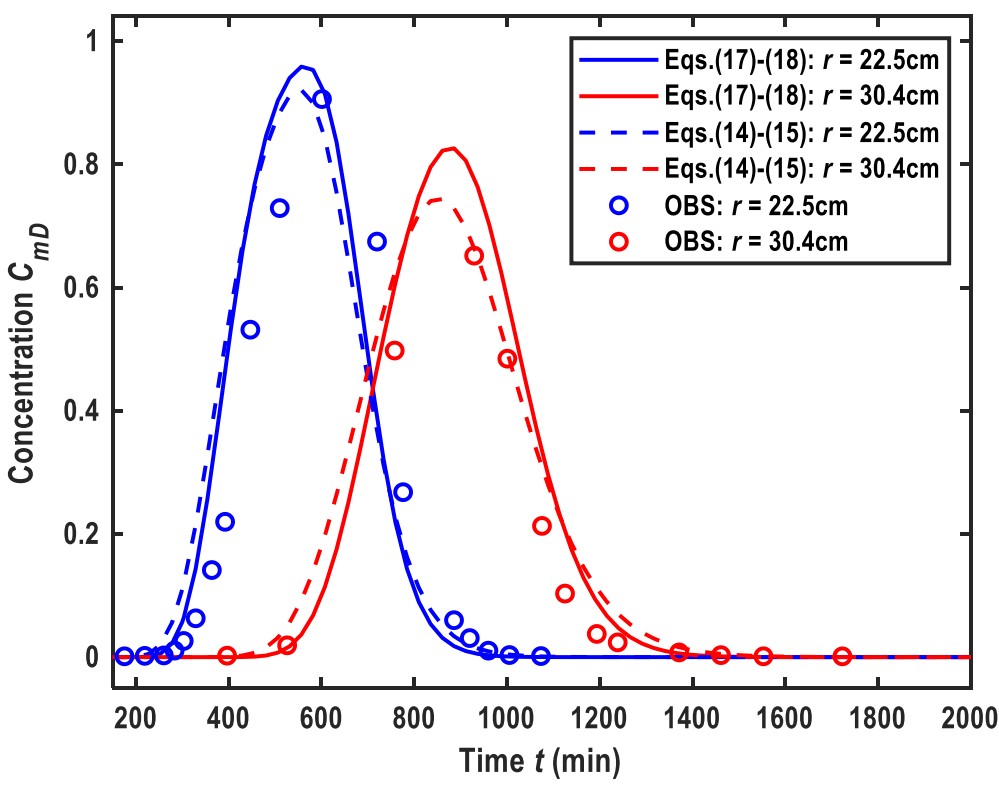

**Figure 3.** Fitness of observed BTC by new solutions of this study, where Eqs. (14) - (15): without scale

effect and Eqs. (17) - (18): with scale effect, respectively.





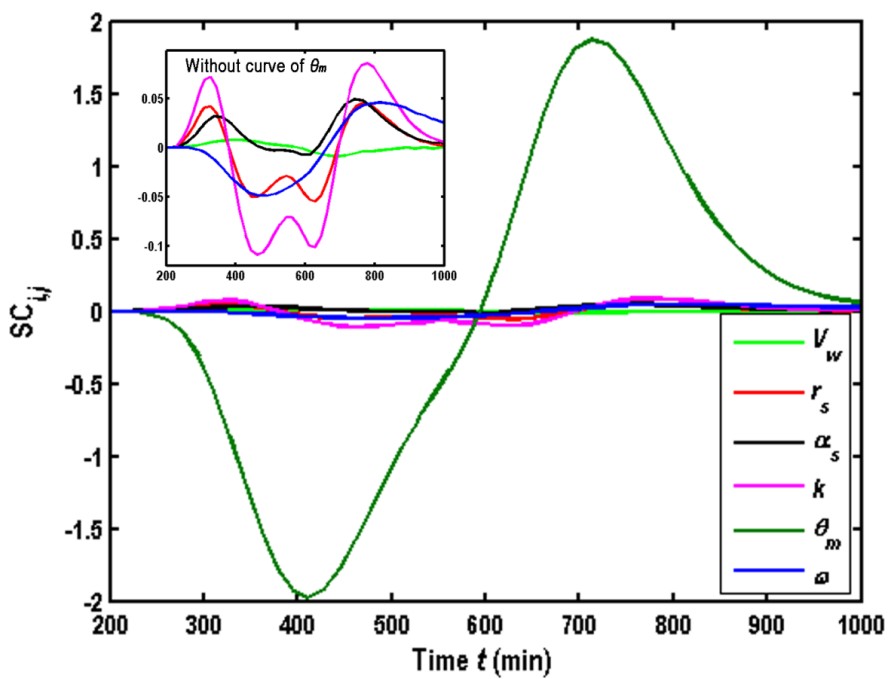

(a). $r$=22.5 cm

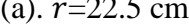

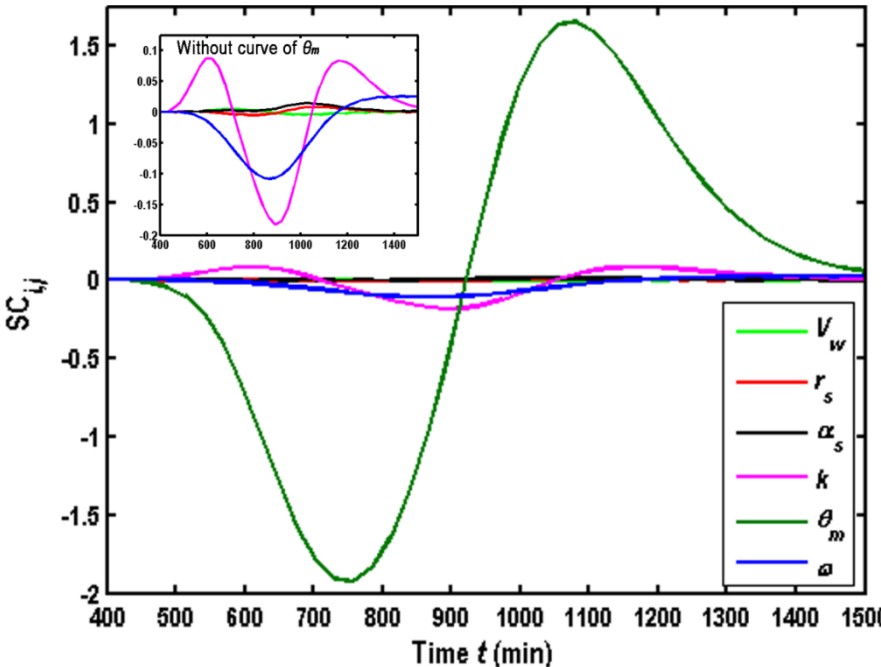


(b). $r$=30.4 cm





**Figure 4.** $SC_{i,j}$ of the parameters $r_w$, $r_s$, $k$, $\theta_m$ and $\omega$ using the parameters estimated by best fitting the experimental data. (a). $r$=22.5 cm, (b). $r$=30.4 cm.

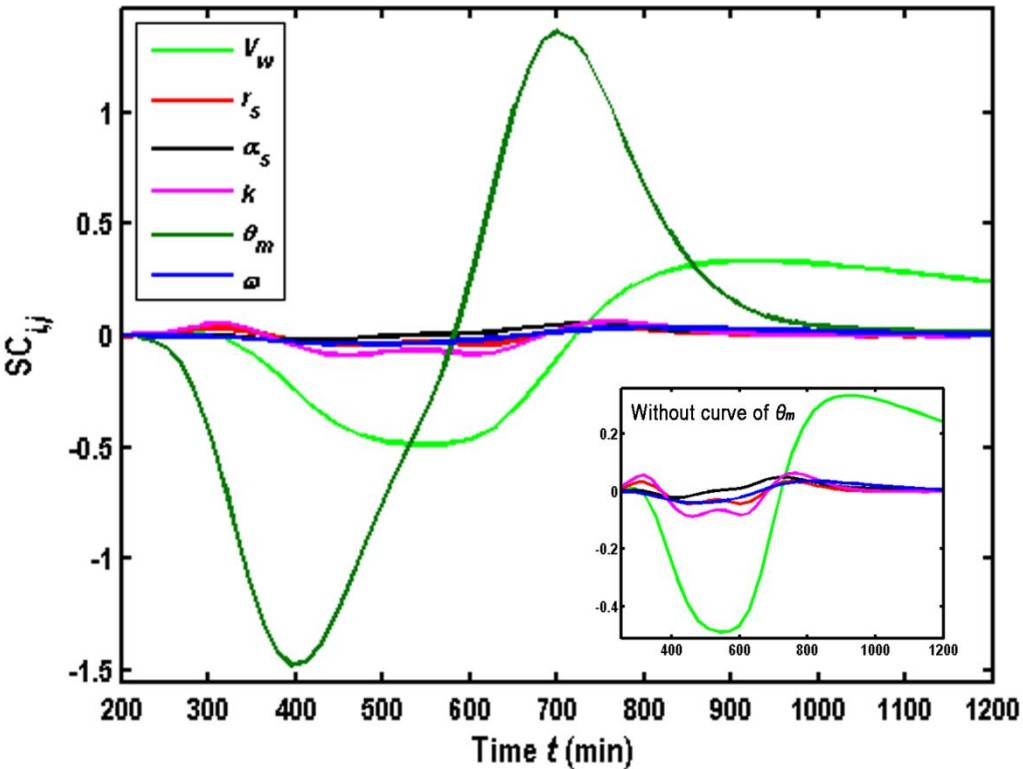

**Figure 5.** $SC_{i,j}$ of the parameters $V_w$, $r_s$, $k$, $\theta_m$ and $\omega$ when increasing $V_w$.





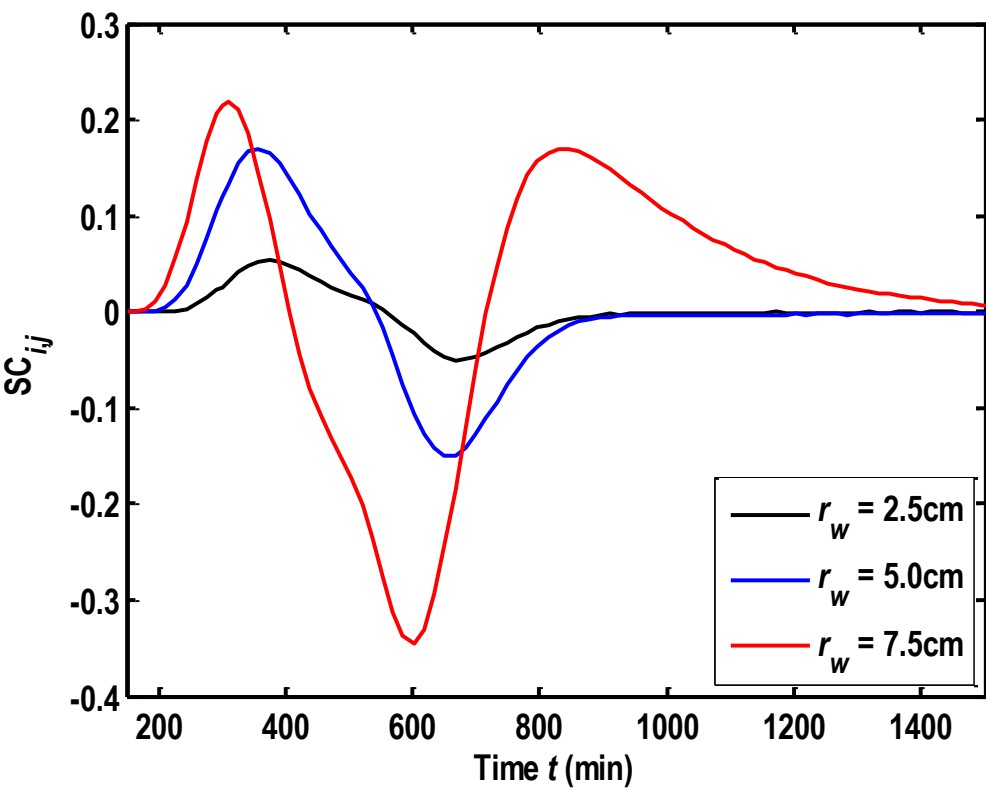


**Figure 6.** $SC_{i,j}$ of $V_w$ for different $r_w$ at $r=22.5$ cm.





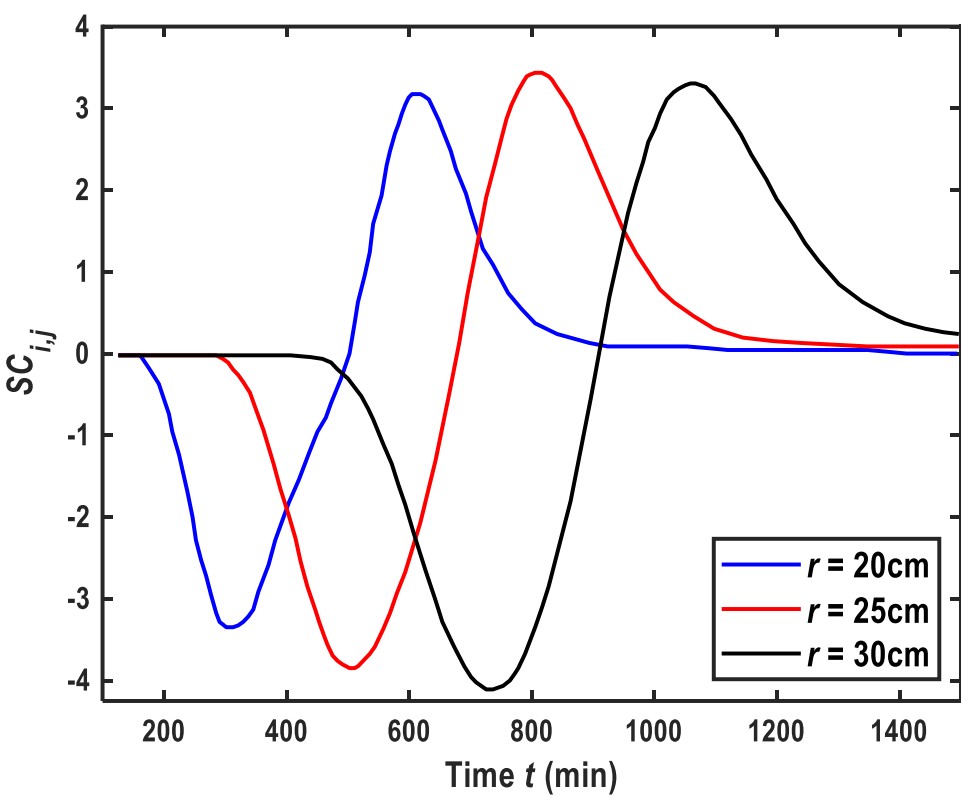

**Figure 7.** $SC_{i,j}$ of $V_w$ for different observed locations when $r_w$=5.0 cm.





**Table Captions**

**Table 1.** Summary of the current models for the radial dispersion around the recharge well.

| Authors | Conceptual models | GE | ME | SCE | SKE | Method |
|---|---|---|---|---|---|---|
| Hoopes and Harleman (1967) | Confined aquifer | ADE | N | N | N | Approximated solution and finite-difference solution |
| Gelbar and Collins (1971) | Confined aquifer | ADE | N | N | N | A boundary layer approximation |
| Tang and Babu (1979), Moench and Ogata (1981), Hsieh (1986), Tang and Peaceman (1987), Yates (1988), Cihan and Tyner (2011), Chen et al. (2012a) | Confined aquifer | ADE | N | N | N | Laplace transform |
| Chen (1985), Chen (1991) | Leaky-confined aquifer | ADE | N | N | N | Laplace transform |
| Chen (1986) | Fracture aquifer | ADE | N | N | N | Laplace transform |
| Falade and Brigham (1989) | Confined aquifer | MIM | N | N | N | Laplace transform |
| Novakowski (1992) | Leaky-confined aquifer | ADE | Y | N | N | Laplace transform |
| Philip (1994) | Confined aquifer | ADE | N | N | N | Finite-difference solution |
| Veling (2001), Veling (2011), Chen et al. (2011) | Confined aquifer | ADE | N | N | N | Generalized Hankel transform |
| Chen et al. (2007), Gao et al. (2009) | Confined aquifer | ADE | N | Y | N | Laplace transform |
| Chen et al. (2012b), Hsieh and Yeh (2014) | Confined aquifer | ADE | N | N | Y | Laplace transform |
| Wang and Zhan (2013) | Leaky-confined aquifer | ADE | N | N | N | Laplace transform |
| Zhou et al. (2017) | Fracture aquifer | MIM | N | N | N | Laplace transform |
| Wang et al. (2018) | Confined aquifer | ADE | Y | N | N | Laplace transform |

Note: "GE", "ME", "SCE", and "SKE" represent governing equation, mixing effect, scale effect, and skin effect, respectively; "Y" and "N" represent whether the effect is considered or not.





**Table 2.** Expressions of coefficients in solutions of Eqs. (14a) - (15b)

| | |
|---|---|
| $N_1$ | $\dfrac{F - H_2 N_2}{H_1}$ |
| $N_2$ | $\dfrac{H_3 H_8 F - H_5 H_6 F}{H_1 H_5 H_7 + H_2 H_3 H_8 - H_2 H_5 H_6 - H_1 H_4 H_8}$ |
| $N_3$ | $\dfrac{H_3 F}{H_1 H_5} - \dfrac{H_2 H_3 N_2}{H_1 H_5} + \dfrac{H_4 N_2}{H_5}$ |
| $H_1$ | $exp\left(\dfrac{r_{wD}}{2\lambda}\right)\left[\dfrac{1}{2}A_i(y_w) - \lambda\left(\dfrac{E_1}{\lambda}\right)^{1/3}A_i'(y_w)\right]$ |
| $H_2$ | $exp\left(\dfrac{r_{wD}}{2\lambda}\right)\left[\dfrac{1}{2}B_i(y_w) - \lambda\left(\dfrac{E_1}{\lambda}\right)^{1/3}exp\left(\dfrac{r_{wD}}{2}\right)B_i'(y_w)\right]$ |
| $H_3$ | $exp\left(\dfrac{r_{sD}}{2\lambda}\right)A_i(y_{1s})$ |
| $H_4$ | $exp\left(\dfrac{r_{sD}}{2\lambda}\right)B_i(y_{1s})$ |
| $H_5$ | $exp\left(\dfrac{r_{sD}}{2}\right)A_i(y_{2s})$ |
| $H_6$ | $exp\left(\dfrac{r_{sD}}{2\lambda}\right)\left[\dfrac{1}{2}A_i(y_{1s}) + \lambda\left(\dfrac{E_1}{\lambda}\right)^{1/3}A_i'(y_{1s})\right]$ |
| $H_7$ | $exp\left(\dfrac{r_{sD}}{2\lambda}\right)\left[\dfrac{1}{2}B_i(y_{1s}) + \lambda\left(\dfrac{E_1}{\lambda}\right)^{1/3}B_i'(y_{1s})\right]$ |
| $H_8$ | $exp\left(\dfrac{r_{sD}}{2}\right)\left[\dfrac{1}{2}A_i(y_{2s}) + (E_2)^{1/3}A_i'(y_{2s})\right]$ |
| $F$ | $F = C_{inj,D}\dfrac{1 - exp(-t_{inj,D}s)}{s} + C_{cha,D}\dfrac{exp(-t_{inj,D}s)}{s}$ |






**Table 3.** Expressions of coefficients in solutions of Eqs. (17a) - (18c)

| | |
|---|---|
| $\mathcal{T}_1$ | $\dfrac{F - W_2\mathcal{T}_2}{W_1}$ |
| $\mathcal{T}_2$ | $\dfrac{W_1 W_5}{W_1 W_4 - W_2 W_3}\mathcal{T}_3 + \dfrac{W_1 W_6}{W_1 W_4 - W_2 W_3}\mathcal{T}_4 - \dfrac{W_3 F}{W_1 W_4 - W_2 W_3}$ |
| $\mathcal{T}_3$ | $\dfrac{W_{13} W_{15} - W_{12} W_{16}}{W_{11} W_{16} - W_{13} W_{14}}\mathcal{T}_4$ |
| $\mathcal{T}_4$ | $\dfrac{W_3 F(W_1 W_8 - W_2 W_7) - W_7 F(W_1 W_4 - W_2 W_3)}{(W_1 W_5 \Theta + W_1 W_6)(W_1 W_8 - W_2 W_7) - (W_1 W_9 \Theta - W_1 W_{10})(W_1 W_4 - W_2 W_3)}$ |
| $\mathcal{T}_5$ | $\dfrac{W_{14}}{W_{16}}\mathcal{T}_3 + \dfrac{W_{15}}{W_{16}}\mathcal{T}_4$ |
| $\mathcal{T}_6$ | $0$ |
| $\Theta$ | $\dfrac{W_{13} W_{15} - W_{12} W_{16}}{W_{11} W_{16} - W_{13} W_{14}}$ |
| $W_1$ | $exp\left(\dfrac{r_{wD}}{2\lambda}\right)\left[\dfrac{1}{2}A_i(y_w) - \lambda\left(\dfrac{E_1}{\lambda}\right)^{1/3}A_i'(y_w)\right]$ |
| $W_2$ | $exp\left(\dfrac{r_{wD}}{2\lambda}\right)\left[\dfrac{1}{2}B_i(y_w) - \lambda\left(\dfrac{E_1}{\lambda}\right)^{1/3}exp\left(\dfrac{r_{wD}}{2}\right)B_i'(y_w)\right]$ |
| $W_3$ | $exp\left(\dfrac{r_{sD}}{2\lambda}\right)A_i(y_{1s})$ |
| $W_4$ | $ex\,p\left(\dfrac{r_{sD}}{2\lambda}\right)B_i(y_{1s})$ |
| $W_5$ | $r_{sD}^m K_m(\varepsilon_1 r_{sD})$ |
| $W_6$ | $r_D^m I_m(\varepsilon_1 r_D)$ |
| $W_7$ | $exp\left(\dfrac{r_{sD}}{2\lambda}\right)\left[\dfrac{1}{2}A_i(y_{1s}) + \lambda\left(\dfrac{E_1}{\lambda}\right)^{1/3}A_i'(y_{1s})\right]$ |
| $W_8$ | $exp\left(\dfrac{r_{sD}}{2\lambda}\right)\left[\dfrac{1}{2}B_i(y_{1s}) + \lambda\left(\dfrac{E_1}{\lambda}\right)^{1/3}B_i'(y_{1s})\right]$ |
| $W_9$ | $-k\varepsilon_1 r_{sD}^{m+1} K_{m-1}(\varepsilon_1 r_{sD})$ |
| $W_{10}$ | $k\{m r_{sD}^{m-1} I_m(\varepsilon_1 r_D) + 0.5\varepsilon_1 r_{sD}^m[I_{m-1}(\varepsilon_1 r_D) + I_{m+1}(\varepsilon_1 r_D)]\}$ |
| $W_{11}$ | $-k\varepsilon_1 r_{0D}^{m+2} K_{m-1}(\varepsilon_1 r_{0D})$ |
| $W_{12}$ | $k\{m r_{0D}^m I_m(\varepsilon_1 r_{0D}) + 0.5\varepsilon_1 r_{0D}^{m+1}[I_{m-1}(\varepsilon_1 r_{0D}) + I_{m+1}(\varepsilon_1 r_{0D})]\}$ |
| $W_{13}$ | $0.5 exp\left(\dfrac{r_D}{2}\right)A_i(y_4) + \varepsilon_1^{1/3}exp\left(\dfrac{r_D}{2}\right)A_i'(y_4)$ |
| $W_{14}$ | $r_{0D}^m K_m(\varepsilon_1 r_{0D})$ |
| $W_{15}$ | $r_{0D}^m I_m(\varepsilon_1 r_{0D})$ |



| $W_{16}$ | $exp\left(\frac{r_{0D}}{2}\right)A_i(y_4)$ |
|---|---|





**Table 4.** Expressions of coefficients in solutions of Eqs. (20a) - (23c).

| | |
|---|---|
| $T_1$ | $\dfrac{F - G_2 T_2}{G_1}$ |
| $T_2$ | $\dfrac{G_3 G_8 F - G_5 G_6 F}{G_1 G_5 G_7 + G_2 G_3 G_8 - G_2 G_5 G_6 - G_1 G_4 G_8}$ |
| $T_3$ | $T_3 = \dfrac{G_3 F}{G_1 G_5} - \dfrac{G_2 G_3 T_2}{G_1 G_5} + \dfrac{G_4 T_2}{G_5}$ |
| $G_1$ | $exp\left(\dfrac{r_{wD}}{2\lambda}\right)\left[\dfrac{1}{2}A_i(\varphi_w) - \lambda\left(\dfrac{E_3}{\lambda}\right)^{1/3}A_i'(\varphi_w)\right]$ |
| $G_2$ | $exp\left(\dfrac{r_{wD}}{2\lambda}\right)\left[\dfrac{1}{2}B_i(\varphi_w) - \lambda\left(\dfrac{E_3}{\lambda}\right)^{1/3}B_i'(\varphi_w)\right]$ |
| $G_3$ | $exp\left(\dfrac{r_{sD}}{2\lambda}\right)A_i(\varphi_{1s})$ |
| $G_4$ | $exp\left(\dfrac{r_{sD}}{2\lambda}\right)B_i(\varphi_{1s})$ |
| $G_5$ | $exp\left(\dfrac{r_{sD}}{2}\right)A_i(\varphi_{2s})$ |
| $G_6$ | $exp\left(\dfrac{r_{sD}}{2\lambda}\right)\left[\dfrac{1}{2}A_i(\varphi_{1s}) + \lambda\left(\dfrac{E_3}{\lambda}\right)^{1/3}A_i'(\varphi_{1s})\right]$ |
| $G_7$ | $exp\left(\dfrac{r_{sD}}{2\lambda}\right)\left[\dfrac{1}{2}B_i(\varphi_{1s}) + \lambda\left(\dfrac{E_3}{\lambda}\right)^{1/3}B_i'(\varphi_{1s})\right]$ |
| $G_8$ | $exp\left(\dfrac{r_{sD}}{2}\right)\left[\dfrac{1}{2}A_i(\varphi_{2s}) + E_4^{1/3}A_i'(\varphi_{2s})\right]$ |
| $F$ | $C_{inj,D}\dfrac{1 - exp(-t_{inj,D}s)}{s} + C_{cha,D}\dfrac{exp(-t_{inj,D}s)}{s}$ |
| $E_3$ | $s + \varepsilon_{m1} + \mu_{m1D} - \dfrac{\varepsilon_{m1}\varepsilon_{im1}}{s + \mu_{im1D} + \varepsilon_{im1}} - \dfrac{a_2\theta_{um}\alpha_2^2 D_u}{2A\theta_{m1}b^2} + \dfrac{b_1\theta_{lm}\alpha_2^2 D_l}{2Ab^2\theta_{m1}}$ |
| $E_4$ | $\dfrac{1}{\eta}\left(s + \varepsilon_{m2} + \mu_{m2D} - \dfrac{\varepsilon_{m2}\varepsilon_{im2}}{s + \mu_{im2D} + \varepsilon_{im2}} - \dfrac{a_2\theta_{um}\alpha_2^2 D_u}{2A\theta_{m2}b^2} + \dfrac{b_1\theta_{lm}\alpha_2^2 D_l}{2Ab^2\theta_{m2}}\right)$ |
| $a_2$ | $-\sqrt{s + \varepsilon_{um} + \mu_{umD} - \dfrac{\varepsilon_{um}\varepsilon_{uim}}{s + \mu_{uimD} + \varepsilon_{uim}}}$ |
| $b_1$ | $\sqrt{s + \varepsilon_{lm} + \mu_{lmD} - \dfrac{\varepsilon_{lm}\varepsilon_{lim}}{s + \mu_{limD} + \varepsilon_{lim}}}$ |





**Table 5.** Parameter values used in Figures 2 and 3

| Parameters | SDM of Chen (2007) | CDM of Chen (2007) | Eqs. (17) - (18) | Eqs. (14) - (15) |
|---|---|---|---|---|
| $\theta$ (-) | 0.58 | 0.58 | \ | \ |
| $\theta_{m1} = \theta_{m2}$ (-) | \ | \ | 0.38 | 0.39 |
| $\theta_{im1} = \theta_{im2}$ (-) | \ | \ | 0.04 | 0.02 |
| $\alpha_1 = \alpha_2$ (cm) | \ | 0.45 | 0.50 | 0.45 |
| $k$ (-) | $2.4 \times 10^{-2}$ | \ | $1.3 \times 10^{-2}$ | \ |
| $r_0$ (cm) | \ | \ | 10000 | \ |
| $\alpha_0$ (cm) | \ | \ | 0.50 | \ |
| $\omega_1 = \omega_2$ (min$^{-1}$) | \ | \ | $1.0 \times 10^{-4}$ | $1.0 \times 10^{-4}$ |
| $\mu_{m1}$ (min$^{-1}$) | 0.0 | 0.0 | $1.0 \times 10^{-7}$ | $1.0 \times 10^{-7}$ |
| $\mu_{m2}$ (min$^{-1}$) | 0.0 | 0.0 | $1.0 \times 10^{-7}$ | $1.0 \times 10^{-7}$ |
| $\mu_{im1}$ (min$^{-1}$) | 0.0 | 0.0 | $1.0 \times 10^{-7}$ | $1.0 \times 10^{-7}$ |
| $\mu_{im2}$ (min$^{-1}$) | 0.0 | 0.0 | $1.0 \times 10^{-7}$ | $1.0 \times 10^{-7}$ |
| $h_{w,inj}$ (cm) | \ | \ | 6.35 | 6.35 |
| $h_{w,cha}$ (cm) | \ | \ | 6.35 | 6.35 |
| $r_s$ (cm) | \ | \ | $r_w$ | $r_w$ |
| $R = R_{m1} = R_{im1} = R_{m2} = R_{im2}$ (-) | 1 | | | |

Note: "SDM" represents the scale-dependent dispersivity model; "CDM" represents the constant dispersivity model; "-" represents that the variable is dimensionless; "\" represents that the variable is not included in the model.



**Table 6.** Errors between observed and computed BTCs in Figure 2.

| Models | Solutions | Observation location (cm) | Error | |
|---|---|---|---|---|
| CDM | Chen et al. (2007) | 22.5 | 0.06 | 0.89 |
| | | 30.4 | 0.83 | |
| | This study | 22.5 | 0.34 | 0.39 |
| | | 30.4 | 0.05 | |
| SDM | Chen et al. (2007) | 22.5 | 0.07 | 0.78 |
| | | 30.4 | 0.71 | |
| | This study | 22.5 | 0.23 | 0.25 |
| | | 30.4 | 0.02 | |