# Peer review of "A General Model of Radial Dispersion with Wellbore Mixing and Skin Effects"

_Hydrology and Earth System Sciences, 2022_

## Author Comment (AC1)

**CHINA UNIVERSITY OF GEOSCIENCES**
SCHOOL OF ENVIROMENTAL STUDIES
WUHAN, HUBEI, CHINA 430074

Dr. Quanrong Wang, Endowed CUG Scholar in Hydrogeology
Tel: +86 15927169156
Email: wangqr@cug.edu.cn

January 30, 2023

Memorandum

To: Dr. Alberto Guadagnini, Editor of Hydrology and Earth System Sciences

Subject: Revision of Paper # hess-2022-372
* * *
Dear Editor:

Upon the recommendation, we have carefully revised Paper # hess-2022-372 entitled "A General Model of
Radial Dispersion with Wellbore Mixing and Skin Effect" after considering all the comments made by the
reviewers. The following is the point-point response to all the comments.

**Response to Reviewer #1:**

**1. Scientific Significance**

The manuscript presents a model for radial dispersion of solutes injected in wells considering the effect of
mixing in the wellbore, the influence of the surrounding skin zone, as well as mobile and immobile regions.
The latter, conceptually modeled as two continuums with spatially uniform parameters which co-exist over
the entire aquifer, allow simulations of early arrivals and long tailing of the breakthrough curves specific to
spatially heterogeneous aquifers. The first-order reactive transport is governed by a system of coupled
equations with constant coefficients which can be solved analytically. The analytical solutions derived by
the authors in Laplace domain are tested against finite-element numerical solutions and experimental data.
It is shown that the new model performs better than partial modes which do not consider simultaneously the
mixing, skin, and heterogeneity effects. As an overall evaluation, the manuscript contributes to the scientific
progress in the research filed and within the scope of the HESS journal.

**Reply:** Thanks a lot. We have carefully revised Paper # hess-2022-372.

**2. Scientific Quality**

(1). The authors present only the solutions in Laplace domain. At page 14 it is mentioned that "the de Hoog
method will be employed to conduct the inverse Laplace transform". A section in the Supplementary
Materials with the computation of the inverse Laplace transform or, at least, references for the method and
the software used in their study should be included.

**Reply:** Implemented. Relevant references have been added. See Lines 283-287.

From Eqs. (14) - (15), one may find that it is not easy to analytically invert the Laplace-domain solution to
obtain the real-time solution. Alternatively, numerical Laplace transform techniques such as the Fourier
series method (Dubner and Abate, 1968), Zakian method (Zakian, 1969), Schapery method (Schapery,

1962), de Hoog method (De Hoog et al., 1982) Stehfest method (Stehfest and Harald, 1970)are called in,
where the de Hoog and Stehfest methods perform better for problems related to radial dispersion (Wang
and Zhan, 2015).

(2). At page 22 it is mentioned the "genetic algorithm (GA) … employed to search the optimal parameter
values", again without any details in Supplementary Materials or references for the algorithm and codes
used. These should be included as well.

**Reply:** Implemented. Relevant references have been added. See Lines 441-446.

In this study, the genetic algorithm (GA) is employed to search the optimal parameter values, such as $\theta_{m2}$,
$\alpha_1$ and $\omega_1$ for CDM of Eqs. (14) - (15), and $\theta_{m2}$, $\alpha_0$, $k$ and $\omega_1$ for CDM of Eqs. (17) - (18). GA is a
stochastic search method, based on natural selection, and it is preferred, due to its efficiency, simple
programmability, and robustness. The GA could be implemented straightforwardly in MATLAB to facilitate
computation (Katoch et al., 2020;Whitley, 1994;Deb et al., 2002).

(3). Apart from these missing details, the applied methods are valid and the results are discussed in with
consideration of related work.

**Reply:** Implemented. The missing details have been added. See Lines 283-287 and 441-446.

**3. Presentation Quality**

The results and conclusions are presented in a clear way and in a good English language. The figures and
tables included are appropriate and the manuscript contains the relevant references to the literature.

**Reply:** Thanks a lot.

**Response to Reviewer #2:**

**1.** The paper proposes a new analytical solution based on the mobile-immobile framework for redial dispersion within a wellbore that considers mixing effect, skin effect, scale effect, aquitard effect and limited media heterogeneity, as this is considered only in the context of the mobile-immobile as a ratio of conductivities in the aquifer, and not as a spatially varying heterogeneity which is a more realistic pattern. The paper is hard to follow, and generally lacks real clarity, specifically there is no in-depth explanation on the "skin-effect" as they previously did in [Li et al., 2019], and it is hard to understand how the derivation differ from their [Wang et al., 2020] paper which focuses on the transport. Moreover, it is not clear how the model is better than existing models? In line 100 the authors claim that other models, namely MRMT, CTRW, and fADE, are "usually unavailable or difficult to develop" yet a quick search show that there are models that cope with that problem well in CTRW [Dentz et al., 2015; Hansen et al., 2016], fADE [Chen et al., 2017; Soltanpour Moghadam et al., 2022], and even a combination of MRMT and CTRW [Kang et al., 2015]. Also, specifically for reactive transport in radial conditions there are experimental evidence for the scaling of dispersion, mixing, and reaction [Edery et al., 2015; Leitão et al., 1996], which are similar to the scaling in this study. The authors should refer to this literature and explain how their analytical solution differ and why is it better as they claim.

**Reply:** Thanks a lot. We have carefully revised Paper # 2021WR030815. This comment is divided into the following questions for response:

**(1)** The paper proposes a new analytical solution based on the mobile-immobile framework for redial dispersion within a wellbore that considers mixing effect, skin effect, scale effect, aquitard effect and limited media heterogeneity, as this is considered only in the context of the mobile-immobile as a ratio of conductivities in the aquifer, and not as a spatially varying heterogeneity which is a more realistic pattern.

**Reply:** The treatment that media heterogeneity effect is described by MIM might be oversimplified for most cases in reality, while they are inevitable for the derivation of the analytical solution. For a heterogeneity aquifer, the solution presented here may be regarded as an ensemble-averaged approximation if the heterogeneity is spatially stationary. If the heterogeneity is spatially non-stationary, then one can apply non-stationary stochastic approach and/or Monte Carlo simulations to deal with the issue, which is out of the scope of this investigation.

**(2)** The paper is hard to follow, and generally lacks real clarity, specifically there is no in depth explanation on the "skin-effect" as they previously did in [Li et al., 2019].

**Reply:** Implemented. More detailed information about skin effects and relevant references have been added. See Lines 63-66 and 80-99.

The skin zone refers to the disturbed region around the well caused by drilling and construction practices or well completion (Yeh and Chang, 2013;Chen et al., 2012; Li et al., 2020;Li et al., 2019;Huang et al., 2019). It is spatially between well screen and aquifer formation zone.

Comparing with aquifer formation zone of interest, the dimension of the skin zone is much smaller, e.g., ranging from 0.1 m to several meters, and it is ignored or included in wellbore. In another word, the effect of the skin zone on radial dispersion (named as skin effect) was negligible. However, numerous previous studies demonstrated that the existence of a skin zone might significantly alter the mechanism of groundwater flow and solute transport around well (Chen et al., 2012;Hsieh and Yeh, 2014;Yeh and Chang, 2013; Li et al., 2020;Li et al., 2019). This is because the physical properties (such as permeability, porosity, dispersivity, and so on) of the skin zone are often vastly different from their counterparts of the formation zone. Previously, studies on the skin effect were mainly concentrated on the groundwater flow process
around the well, and much less attention was paid to solute transport processes. To date, few studies
considered the skin effect among the above-mentioned analytical models on radial dispersion, such as
Chen et al. (2012), Hsieh and Yeh (2014), Huang et al. (2019) and Li et al. (2020). Chen et al. (2012)
proposed an analytical solution of solute transport with skin effect to investigate the influences of
dispersivity on radial dispersion, soon after, Hsieh and Yeh (2014) extended the model of Chen et al. (2012)
by taking into account a third-type (Robin) condition. Huang et al. (2019) demonstrated that the skin effect
has a major influence on observed breakthrough curves (BTCs) for radially convergent tracer tests.
Recently, Li et al. (2020) developed the analytical model for radial reactive transport with skin effect to
investigate the impacts of dispersivity, effective porosity and mass transfer coefficient in skin zone on radial
dispersion. The above-mentioned studies demonstrated the skin effects are significant for radial dispersion.

**(3)** and it is hard to understand how the derivation differ from their [Wang et al., 2020] paper which focuses
on the transport.

**Reply:** Implemented. See Lines 116-118.

Wang et al. (2020) developed a four-stage radial dispersion model with aquitard and wellbore mixing
effects under the MIM framework; however, the skin and scale effects were ignored in Wang's model,
which were considered in this study. The methodology between these two papers is also different. In Wang
et al. (2020), Laplace transform and Green's function methods are used to derive the analytical solution,
while only Laplace transform method is used in this study.

**(4)** Moreover, it is not clear how the model is better than existing models? In line 100 the authors claim that
other models, namely MRMT, CTRW, and fADE, are "usually unavailable or difficult to develop" yet a quick
search show that there are models that cope with that problem well in CTRW [Dentz et al., 2015; Hansen et
al., 2016], fADE [Chen et al., 2017; Soltanpour Moghadam et al., 2022], and even a combination of MRMT
and CTRW [Kang et al., 2015]. Also, specifically for reactive transport in radial conditions there are
experimental evidence for the scaling of dispersion, mixing, and reaction [Edery et al., 2015; Leitão et al.,
1996], which are similar to the scaling in this study. The authors should refer to this literature and explain
how their analytical solution differ and why is it better as they claim.

**Reply:** Implemented. The sentence of 'the analytical solutions associated with radial dispersion are usually
unavailable or difficult to develop' in the original manuscript has been deleted, and relevant references
have been added. See Lines 103-123.

As for reactive transport in heterogeneous media, the BTCs may exhibit a host of non-Fickian
characteristics such as early arrival and heavy tailing (Di Dato et al., 2017;Molinari et al., 2015).
Alternatively, many non-Fickian transport models have been developed, such as the multi-rate mass
transfer model (MRMT) (Le Borgne and Gouze, 2008;Haggerty et al., 2001), mobile-immobile model (MIM)
(van Genuchten and Wierenga, 1976;Zhou et al., 2017;Wang et al., 2020), continuous-time random-walk
models (CTRW) (Dentz et al., 2015;Hansen et al., 2016), fractional-derivative ADE models (fADE)
(Soltanpour Moghadam et al., 2022;Chen et al., 2017), a combination of MRMT and CTRW (Kang et al.,
2015), and so on (Zheng et al., 2019;Lu et al., 2018). Although the models of MRMT, CTRW and fADE
perform well in modeling non-Fickian transport, it is not easy to obtain the analytical solutions of these
models. Meanwhile, these theories are usually not easy to apply for solving regional-scale transport
problems, as pointed out in a recent study (Zheng et al., 2019). MIM is an extension of ADE by considering
both flowing and stagnant regions in porous media and mass transfer between them (van Genuchten and
Wierenga, 1976;Zhou et al., 2017;Wang et al., 2020), Zhou et al. (2017) and Wang et al. (2020) derived the
MIM solutions of radial dispersion. However, the skin effect and the scale effect were ignored in their studies, which will be investigated in this study. Besides the MRMT, MIM, CTRW, and fADE models,
another approach to represent the heterogeneity is to use a scale-dependent dispersivity (or dispersion) in
the ADE or MIM models (Haddad et al., 2015;Gelhar et al., 1992). Gao et al. (2009a) and Chen et al. (2007)
discussed radial dispersion and found that the scale-dependent dispersion effect was not negligible. There
are also experimental evidence for the scaling of dispersion, mixing, and reaction (Leitão et al., 1996;Edery
et al., 2015).

**2.** Line 135-137 needs to be clarified

**Reply:** Implemented. See Lines 151-159.

In this study, we mainly focus on developing analytical solutions of radial dispersion with a Heaviside step
source (or step function for abbreviation hereinafter), as solutions of a variety of injection scenarios can be
easily obtained on the basis of such a step source solution, as shown in Eq. (A2) in Supplementary
Materials, Eqs. (4a) - (4b), or Eqs. (5a) - (5b). Assuming that $t_{inj}$ is the duration of the step source, the
solute source concentration ($C_0$) is $C_{inj}(t)$ when time is smaller than $t_{inj}$, while it is $C_{cha}(t)$ when time is
greater than $t_{inj}$, in which $C_{inj}(t)$ and $C_{cha}(t)$ represent the solute concentrations [ML$^{-3}$] in the wellbore
before time $t_{inj}$ and after time $t_{inj}$, respectively; When $C_{cha}(t) = 0$ and $t_{inj}$ approaches zero but the
total injected mass remains finite, the model of the step source reduces to the model of the instantaneous
injection.

**3.** Are we defining the asymptotical value for the model in line 137-139, please clarify.

**Reply:** Implemented. The value of $t_{inj}$ ($t_{inj} = 300$ mim) has been added in Table 5. See Table 5.

**4.** Line 157-162 defines reaction rate (or radioactive decay, or biodegradation), and retardation factor yet
there is no example to using these parameters in the results since R=1, μ is so small it is negligeable, so
the sensitivity to these parameters must be small. Can the author comment on the choice of parameters?
Also, why is this part in the supplementary and not in the text?

**Reply:** Implemented. The parameter selection has been added (See Lines 392-402), and the sensitivity of
results to $R$ and $\mu$ could be seen in Figures 4 and 5.

The parameters used in the numerical simulation are: $r_w = 2.5$ cm; $r_s = 12.5$ cm; $Q_{inj} = Q_{cha} =$
$100$ ml/s; $t_{inj} = 300$ s; $\alpha_1 = 2.5$ cm; $\alpha_2 = 2.5$ cm; $\theta_m = 0.30$; $\theta_{im} = 0.01$; $\omega = 0.001$ d$^{-1}$; $R_{m1} =$
$R_{im1} = R_{m2} = R_{im2} = 1$ ; $B = 50$ cm; $\mu_{m1} = \mu_{m2} = \mu_{im1} = \mu_{im2} = 10^{-7}$ s$^{-1}$, and $h_{w,inj} =$
$h_{w,cha} = B$. These parameters are from the experimental applications of Chao (1999), Chen et al. (2017),
Wang et al. (2018) and Wang et al. (2020), in which Wang et al. (2020) summarized the values of reaction
rate, retardation factor, dispersivity, porosity, and first-order mass transfer coefficient for sandy and clay
used in numerous investigations, as shown in Table 4 of Wang et al. (2020). In addition, the values of
retardation factor and reaction rate represent that the chemical reaction and sorption are weak for the
tracer of KBr in the experiment of Chao (1999). It is not surprising since KBr is commonly treated as a
"conservative" tracer.

**5.** As the COMSOL solution was based on equation 14, which is the basis for the analytical solution
equation 20-23, it is not surprising that the match between them in figure S4 is good, yet why do they differ so much from the observation in Chao et al 1999? Moreover, can the authors supply an R-square or
quantify how well the analytical solution performs for all figure, and not just figure 3, where the COMSOL
solution is very different? Please, add the error to the figure caption as it is confusing to switch between the
figure to the table?

**Reply:** Implemented. This comment is divided into the following questions for response:

**(1)** As the COMSOL solution was based on equation 14, which is the basis for the analytical solution
equation 20-23, it is not surprising that the match between them in figure S4 is good, yet why do they differ
so much from the observation in Chao et al 1999?

**Reply:** Implemented. The COMSOL solution is a numerical solution, and it is used to test the new
analytical solution of this study. The models used to interpret the observation in Chao et al 1999 are
analytical solutions, not numerical solutions. Meanwhile, New Figures S4a and S4b have been added in
*Supplementary Materials*. Figure S4 in the original *Supplementary Materials* has been changed into Figure
S5. See Lines 403-408, Figures 1 and S4.

**(2)** Moreover, can the authors supply an R-square or quantify how well the analytical solution performs for
all figure, and not just figure 3, where the COMSOL solution is very different? Please, add the error to the
figure caption as it is confusing to switch between the figure to the table?

**Reply:** Implemented. See Lines 452-458, Figures 1 to 3 and Figures S4 and S5, Table 6.

The COMSOL solution is only used to test the accuracy of the new models of this study, as shown in
Figures 1 and S4. The R-square ($R^2$) has been added in Figures 1 and S4 representing fitness between
analytical solution and numerical solution, and the $R^2$ has been added in Figures 2, 3 and S5 representing
fitness between computed and observed BTCs. The error ($E_r$) form Table 6 have been added to Figures 2,
3 and S5.

**6.** Another point is that there is no explanation as to why the error is so big, and why the analytical solution
is better than the numerical one with respect to the error.

**Reply:** Implemented. See Lines 427-433 and Figures 2, 3 and S5.

In this study, the models used to interpret the observation in Chao et al 1999 are analytical solutions, not
numerical solutions.

Figure 2 shows the fitness of observed BTC by the solution of Chen et al. (2007) which considers the scale
effect but ignores the mixing and skin effects. One might find that the fitness between computed and
observed BTCs was obvious. We found that it was probably due to the following two reasons. Firstly, the
model of Chen et al. (2007) used to best fit the data is an instantaneous slug test model, which was a
rather gross approximation of the injection which lasted about 5 hours. A more proper way is to treat the 5
hours injection as a step source. Secondly, the solution of Chen et al. (2007) only considered the scale-
dependent dispersivity, but ignored the mixing effect and the mass transfer between the mobile and
immobile domains.

So, we used the new analytical solution of this study to re-interpret the observed data, as shown in Figures
3 and S5.

**7.** To summarize, the paper seems like an important contribution as it considers many physical aspects for
radial dispersion (reaction, retardation, conductivity change in the skin area), and provides an analytical
solution that considers these aspects. However, at the moment the advantage of the analytical solution,
when compared to experimental data and even to the numerical solution is not clear enough. The paper is
not approachable, as the figures need to be combined with the error while all the details of the modeling
and results need to be ordered and clarify. Lastly, there is a bulk of literature that need to be added to put
this work in the right context. I believe that addressing these comments will make the paper more
approachable, provide the right context and make a stronger case for the analytical solution presented here.

**Reply:** Implemented. We have carefully revised the manuscript after considering all of the above-
mentioned comments. Thanks a lot for such valuable comments.

____________________________________________________________________________

If you have any further questions about this revision, please contact me.

Sincerely Yours,

Quanrong Wang, PhD, PG.

Professor and

Holder of Endowed CUG Scholar in Hydrogeology

---

## Author Response (AR2)

**CHINA UNIVERSITY OF GEOSCIENCES SCHOOL OF ENVIROMENTAL STUDIES WUHAN, HUBEI, CHINA 430074**

Dr. Quanrong Wang, Endowed CUG Scholar in Hydrogeology Tel: +86 15927169156 Email: wangqr@cug.edu.cn

1

March 29, 2023

- 2 Memorandum
- 3 To: Dr. Alberto Guadagnini, Editor of Hydrology and Earth System Sciences
- 4 Subject: Revision of Paper # hess-2022-372
- 5 Dear Editor:
- 6 Upon the recommendation, we have carefully revised Paper # hess-2022-372 entitled "A General Model of
- 7 Radial Dispersion with Wellbore Mixing and Skin Effects" after considering all the comments made by the
- 8 reviewers. The following is the point-point response to all the comments.
- 9

**10 **Response to Reviewer #1:**

- 11 The model of radial dispersion presented in the manuscript is governed by a system of coupled equations
- 12 with constant coefficients. This system has a straightforward analytical solution in Laplace space,
- 13 presented by the authors in Supplementary Materials. The main achievement of the model could be its
- 14 practical feasibility, which requires the computation of the inverse Laplace transform of the solution and an
- 15 optimization procedure to fit the analytical solution and the experimental data. Unfortunately, these two
- 16 parts of the model are still missing in the revised manuscript.
- 17 **Reply:** Thanks a lot. We have carefully revised Paper # hess-2022-372.
- 18 In my previous review, I recommended that the authors provide sufficient details in the Supplementary
- 19 Materials or, if this is the case, references for THE METHOD AND THE SOFTWARE used, for both the
- 20 computation of the inverse Laplace transform and the genetic algorithm employed in the optimization
- 21 procedure. Instead, in their reply the authors enumerate several references, without pointing out a method
- and a software they have used. The absence of these precise details, which allow the reproducibility of the
- results, leads to a lack of trust in the modeling approach presented in the manuscript. Therefore, I reiterate
- 24 the recommendation that the authors provide these essential details that can make their work useful to
- 25 potentially interested readers.
- Reply: Implemented. De Hoog method and GA software and methods have been added. Please see Lines
  287-289 and 444-446.
- 28

**29 Response to Reviewer #2:**

- 30 The authors answered my concerns in a satisfactory way. The paper is clearer and has the right context
- 31 with respect to other available models. I would love to see the error incorporated in the figure and not just in
- 32 the legend but this is a minor thing and I leave it to the editor decision.
- 33 **Reply**: Thanks a lot. The errors have been added to Figures 2 and 3. Please see Figures 2 and 3.

- 34
- 35 If you have any further questions about this revision, please contact me.
- 36 Sincerely Yours,
- 37 Quanrong Wang, PhD, PG.
- 38 Professor and

Quarong Wang

39 Holder of Endowed CUG Scholar in Hydrogeology

---

## Author Response (AR3)

**CHINA UNIVERSITY OF GEOSCIENCES**
SCHOOL OF ENVIROMENTAL STUDIES
WUHAN, HUBEI, CHINA 430074

Dr. Quanrong Wang, Endowed CUG Scholar in Hydrogeology
Tel: +86 15927169156
Email: wangqr@cug.edu.cn

1 April 6, 2023

2 Memorandum

3 To: Dr. Alberto Guadagnini, Editor of Hydrology and Earth System Sciences

4 Subject: Revision of Paper # hess-2022-372
* * *
5 Dear Editor:

6 Upon your recommendation, we have carefully revised Paper # hess-2022-372 entitled "A General Model
7 of Radial Dispersion with Wellbore Mixing and Skin Effects".

8

9 **Response to Editor:**

10 While I do appreciate the prompt reply, the quality of English (also in the newly added sentences) is not
11 sufficient. I would recommend a thorough revision along these lines.

12 **Reply:** Thanks a lot. The English has been polished by two native speckers, Mohamed Hussein and James
13 JS Yeanay Jr, who are mentioned in the Acknowledgements.

14
* * *
15 If you have any further questions about this revision, please contact me.

16 Sincerely Yours,

17 Quanrong Wang, PhD, PG.

18 Professor and

19 Holder of Endowed CUG Scholar in Hydrogeology